# Energy-saving hydrogen production by seawater electrolysis coupling tip-enhanced electric field promoted electrocatalytic sulfion oxidation

Tongtong Li[1], Boran Wang[1], Yu Cao[2], Zhexuan Liu[1], Shaogang Wang [3], Qi Zhang[1], Jie Sun[2] & Guangmin Zhou [1] ✉

Hydrogen production by seawater electrolysis is significantly hindered by high energy costs and undesirable detrimental chlorine chemistry in seawater. In this work, energy-saving hydrogen production is reported by chlorine-free seawater splitting coupling tip-enhanced electric field promoted electrocatalytic sulfion oxidation reaction. We present a bifunctional needle-like $Co_3S_4$ catalyst grown on nickel foam with a unique tip structure that enhances the kinetic rate by improving the current density in the tip region. The assembled hybrid seawater electrolyzer combines thermodynamically favorable sulfion oxidation and cathodic seawater reduction can enable sustainable hydrogen production at a current density of 100 mA cm$^{-2}$ for up to 504 h. The hybrid seawater electrolyzer has the potential for scale-up industrial implementation of hydrogen production by seawater electrolysis, which is promising to achieve high economic efficiency and environmental remediation.

Hydrogen (H$_2$), an attractive renewable energy with the advantages of low pollution and high energy density, is important for future energy transition and reducing reliance on fossil fuels[1,2]. However, traditional H$_2$ production methods suffer from severe environmental pollution, complex equipment processes, high investment operational costs, and excessive energy consumption[3]. In contrast, electrocatalytic water splitting has received widespread attention as a zero-pollution emission and efficient method to produce H$_2$[4,5]. Seawater, which accounts for over 96% of the Earth's total water resources and is widely distributed, can serve as an abundant aqueous electrolyte feedstock for sustainable H$_2$ production. Nonetheless, seawater electrolysis faces two significant barriers. First, the faster kinetics of the chloride ion oxidation reaction (ClOR, Cl$^-$ + 2OH$^-$ → ClO$^-$ + 2H$_2$O + 2e$^-$) competes with the desired anodic oxygen evolution reaction (OER)[6]. Second, the presence of chloride ion (Cl$^-$) in seawater causes corrosion and degradation of the electrocatalysts, thereby affecting hydrogen

production efficiency and electrolyzer lifespan[7–9]. To achieve the commercialization of hydrogen production by seawater electrolysis, the development of efficient electrocatalytic system and corresponding electrocatalysts applicable for seawater electrolysis to lower electrolysis voltage and avoid detrimental chlorine chemistry is necessary.

As the half-reaction of water electrolysis, the anodic OER is a four-electron transfer process with sluggish kinetics, leading to increased overall energy consumption[10]. To address this issue, researchers have proposed a series of anodic alternative reactions with lower energy requirements. Duan's group fabricated a multi-metallic thin-film catalyst that efficiently hydrolyzes polyethylene terephthalate into recyclable monomers while generating hydrogen, providing an effective alternative to OER[4]. Similarly, Wen et al. developed a high-entropy alloy electrocatalyst that catalytically converts glycerol into high-value-added products in alkaline media[7]. It offered alternative reactions to traditional OER, achieving efficient glycerol conversion,

[1]Tsinghua-Berkeley Shenzhen Institute & Tsinghua Shenzhen International Graduate School, Tsinghua University, Shenzhen 518055, PR China. [2]School of Chemical Engineering and Technology, Tianjin University, Tianjin 300072, PR China. [3]Shenyang National Laboratory for Materials Science, Institute of Metal Research, Chinese Academy of Sciences, Shenyang 110016, PR China. ✉e-mail: guangminzhou@sz.tsinghua.edu.cn

hydrogen generation, and reducing energy consumption. Li and co-workers introduced the urea oxidation reaction with a lower thermodynamic potential (370 mV *vs*. RHE) as a substitute for OER in the electrolyzer, achieving energy-efficient hydrogen production and urea-containing sewage treatment[11]. These studies demonstrate that these alternative reactions significantly lower the electrolyzer's energy consumption. Among the alternative anodic oxidation reactions, the sulfion oxidation reaction (SOR, $S^{2-} - 2e^- \rightarrow S$, $E_0 = -480$ mV *vs*. SHE) is considered feasible[12]. Sulfion-containing sewage generated from various processes (natural gas, syngas, refinery gas, mineral decomposition, and sewage treatment processes) poses severe hazards to human health, crops, and plants. The most commonly used method for treating sulfides in natural gas and oil is the Claus method. Efficient electrochemical sulfion oxidation methods for sewage treatment offer high purification efficiency without the need for additional oxidants or complex separation processes, thus achieving environmental protection goals. Furthermore, using thermodynamically favorable SOR instead of OER combined with HER enables energy-efficient $H_2$ production and avoids competition between the ClOR and OER (Supplementary Fig. 1)[13]. Improving the catalytic activity and stability of electrocatalysts for SOR is needed to meet the requirements of practical applications. Nonetheless, current catalysts for SOR face bottleneck constraints in terms of long-term stability at high current densities, which are mainly attributed to the unfavorable mass transfer rates of traditional catalysts resulting in significant sulfion deposition during operation, which passivates catalysts[14,15].

Herein, we synthesized a bifunctional needle-like $Co_3S_4$ ($n$-$Co_3S_4$) catalyst grown on nickel foam (NF) with a local field enhancement effect, for efficient SOR-assisted seawater electrolysis hydrogen production and sulfur recovery. Finite element analysis, $S^{2-}$ adsorption test, and Kelvin probe force microscopy have verified that the tip curvature radii could have a positive effect on the tip-enhanced local electric field in promoting zonal current density and $S^{2-}$ concentration, thereby accelerating the SOR rate. DFT calculations show that needle-like catalyst ($n$-$Co_3S_4$@NF) has a spontaneous adsorption capacity for $S^{2-}$, with the lowest energy barrier (1.40 eV) for the rate-determining step on its (311) facet originating from the oxidation step $S^* \rightarrow S_2^*$, exhibiting the highest efficient polysulfide conversion capacity. By using the $n$-$Co_3S_4$@NF as a bifunctional electrocatalyst for SOR-assisted seawater electrolysis, the hybrid seawater electrolyzer (HSE) exhibits a remarkable durability of 504 h. It has a lower power consumption of 1.2 kWh m$^{-3}$ $H_2$, allowing for more than 67.9% reduction in power consumption compared to conventional alkaline seawater electrolyzer (ASE) system. What's important, the coupled electrolyzer can be stably driven by the temperature difference power generation device and solar cell, offering an important idea for large-scale high-efficiency, sustainable, and low-energy seawater hydrogen production.

## Results
### Catalysts preparation and electric field mechanism investigations
Given the local field enhancement effect of needle-like structures in improving the performance of electrocatalytic reactions, we attempt to apply this structure to the electrocatalytic SOR process[16–18]. The preparation procedures for the needle-like structure $Co_3S_4$ electrode are illustrated in Fig. 1a, b. The needle-like $Co(OH)_2$ precursor was grown on a Ni foam substrate (denoted as $n$-$Co(OH)_2$@NF) via a hydrothermal method. The resulting $n$-$Co(OH)_2$@NF was converted into the needle-like structure $Co_3S_4$ (denoted as $n$-$Co_3S_4$@NF) by vulcanization under the $S^{2-}$ atmosphere. As a comparison, $Co_3S_4$ with a rod-like structure grown on a Ni foam substrate (denoted as $r$-$Co_3S_4$@NF) was also prepared by a similar two-step hydrothermal method. In a typical synthesis, the precursors $n$-$Co(OH)_2$@NF and $r$-$Co(OH)_2$@NF (Supplementary Figs. 2 and 3) were grown on a substrate of Ni foam through a hydrothermal method. The powder X-ray

diffraction (XRD) was carried out to confirm the crystal structure of the as-prepared samples. In the XRD pattern of the $n$-$Co_3S_4$@NF (Supplementary Fig. 4a), the diffraction peaks at 26.3°, 31.5°, 38.4°, 47.1°, 50.3°, and 55.0° are well matched with the spinel-structured $Co_3S_4$ (JCPDS 47-1738), respectively. It proves that the successful transformation of $Co(OH)_2$ into $Co_3S_4$. To be noted, the characteristic peaks located at 44.7°, 52.0°, and 76.6° are attributed to the metallic nickel substrate[19,20]. Scanning electron microscopy (SEM) and transmission electron microscope (TEM) images in Fig. 1c and Supplementary Figs. 5 and 6 show that the $n$-$Co_3S_4$@NF has a length-width ratio of *ca*. 20:1 with a tip structure firmly grown on the Ni foam surface, which can provide more active sites while highlighting the tip field enhancement effect. The $r$-$Co_3S_4$@NF has a length-width ratio of *ca*. 4:1 with a rod-like structure (Fig. 1d and Supplementary Figs. 7 and 8). Notably, the structural characteristics of $n$-$Co_3S_4$@NF and $r$-$Co_3S_4$@NF have been largely preserved after the vulcanization process, which is instructive for the preparation of other types of needle-like metal sulfides. In addition, it is exciting to note that the surface of the resulting $n$-$Co_3S_4$@NF and $r$-$Co_3S_4$@NF became significantly rougher after the vulcanization process, facilitating the exposure of catalytic active sites (Supplementary Figs. 2, 3, 5, and 7). The high-resolution transmission electron microscopy (HRTEM) image of $n$-$Co_3S_4$@NF (Fig. 1e) clearly shows interplanar spacings of 0.284, 0.544, 0.333, and 0.235 nm corresponding to the (311), (111), (220), and (400) crystal planes of the spinel-structured $Co_3S_4$ (JCPDS 47-1738). Energy dispersive X-ray spectroscopy (EDX) verifies the uniform distribution of S and Co elements in $n$-$Co_3S_4$@NF (Fig. 1f). Overall, the two-step hydrothermal method successfully constructed needle-like structure $Co_3S_4$ catalysts with abundant and highly exposed active sites. The chemical states of Co and S elements on the catalyst surface were investigated using X-ray electron diffraction (XPS). As shown in Supplementary Fig. 9a, the high-resolution Co 2$p$ of the $n$-$Co_3S_4$@NF catalyst can be deconvoluted into two pairs of spin-orbit double peaks (Co 2$p_{3/2}$ and Co 2$p_{1/2}$) with satellite peaks (Sat.). The peaks at 798.30 and 782.20 eV correspond to $Co^{2+}$ and the additional two characteristic peaks at 796.70 and 781.46 eV correspond to $Co^{3+}$ [21,22]. The characteristic peak of $Co^{2+}$ exhibits a leftward shift of 0.29 eV relative to the precursor $n$-$Co(OH)_2$@NF. There is a shift caused by the dual contribution of the electronegativity and easy polarizability of the sulfion in $Co_3S_4$, which contributes to the adjustment of the charge distribution of the adjacent metal ions by reinforcing the electron transfer capacity[23]. Noteworthy, the $Co^{2+}$ characteristic peak shift of $n$-$Co_3S_4$@NF is more obvious in comparison with $r$-$Co_3S_4$@NF (Supplementary Figs. 9a and 10a), whereby we speculate that the special needle-like structure can facilitate the improvement of the catalytic ability of the $Co_3S_4$ catalyst. The peaks at binding energies of 168.89, 164.00, and 161.83 eV are attributed to $SO_4^{2-}$, S 2$p_{1/2}$, and S 2$p_{3/2}$, respectively (Supplementary Figs. 9b and 10b)[24,25]. The Raman spectra revealed vibrational peaks that are highly compatible with the crystal structure of $Co_3S_4$, including $A_g$, $E_g$, and $F_{2g}$ (Supplementary Fig. 11)[26]. Their Raman modes reveal obvious broadening and weakening, which is related to the striking discrepancy with morphological structures of $Co_3S_4$[27]. In addition, based on the above analysis, we can determine that apart from the morphological differences, there is no obvious difference in crystal structures and surface chemical states between $n$-$Co_3S_4$@NF and $r$-$Co_3S_4$@NF. The needle-like structure has a typical tip morphology, and we hypothesize with the sharpness of the morphology, the electron density in the tip region will be heavily clustered under the same current density, leading to an increase in the strength of the surrounding electric field. It can enhance the adsorption of surrounding reactive ions (e.g., $S^{2-}$ and $OH^-$), further enhancing the SOR and HER catalytic activities[16]. To demonstrate this speculation, we used a finite element method to explore the prospect of tip-enhanced nanoscale field enhancement and cation concentration, in which a cylinder immersed in solution was used to represent the rod-like

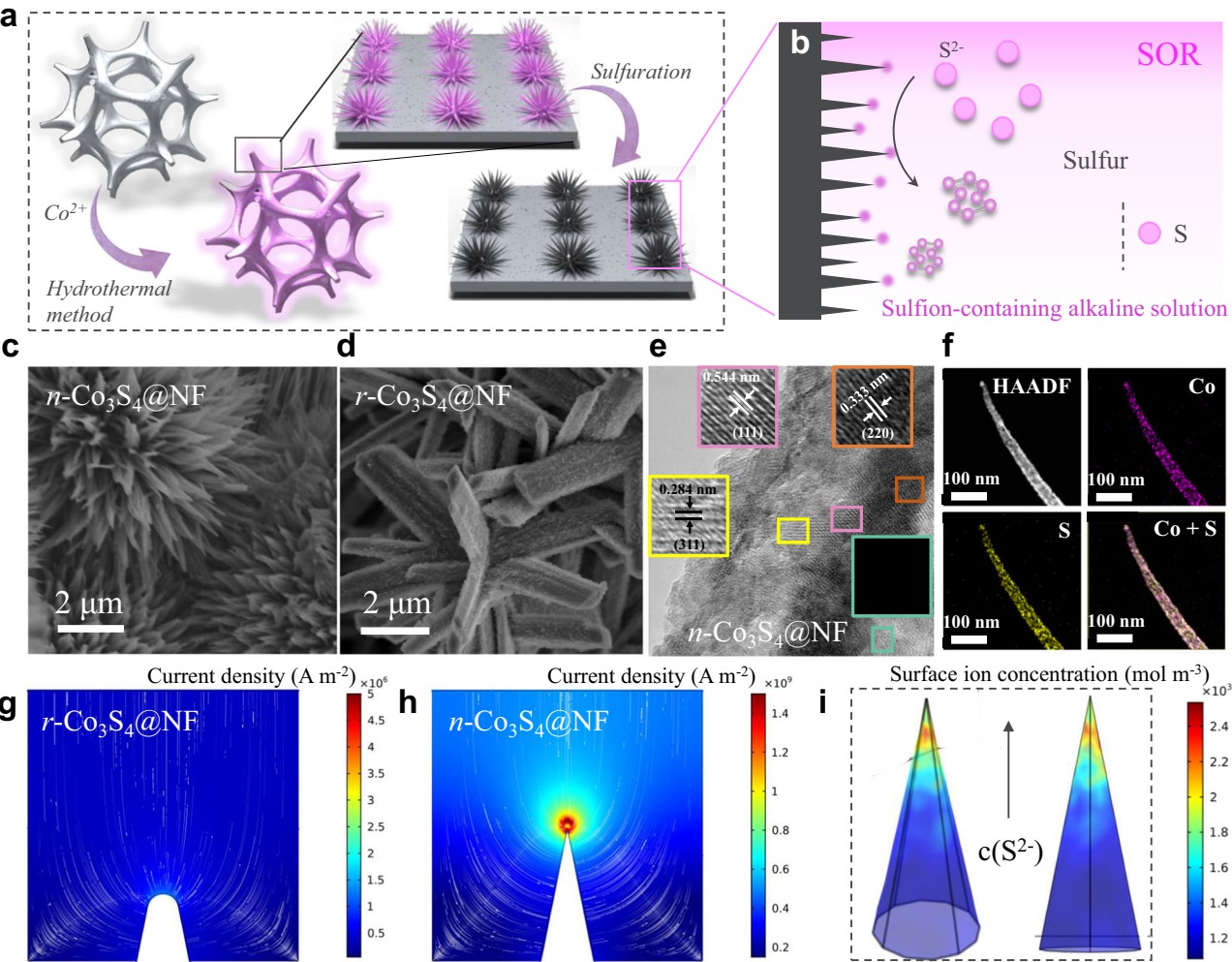

**Fig. 1 | Synthesis, characterization, and electric field enhancement mechanism.**
Schematic illustration of (**a**) the synthesis process for $n$-Co$_3$S$_4$@NF and (**b**) the tip-enhanced electric field effect. SEM images of (**c**) $n$-Co$_3$S$_4$@NF and (**d**) $r$-Co$_3$S$_4$@NF. **e** HRTEM image and (**f**) dark-field TEM image with corresponding elemental mapping images for $n$-Co$_3$S$_4$@NF. Calculated electric field, current density, and S$^{2-}$ near the electrode tip as a function of electrode morphology. The colored plots represent the free electron density distribution on the electrode surface, and the white lines represent the electric field distributions of (**g**) $r$-Co$_3$S$_4$@NF and (**h**) $n$-Co$_3$S$_4$@NF. **i** Relationship between S$^{2-}$ adsorbed on the surface of $n$-Co$_3$S$_4$@NF and current density at tip.

structure (Fig. 1g), while approximate cone-shaped geometries were indicated as a needle-like structure (Fig. 1h)[18]. It can be noticed that the current density in the sharpest region of the tip of the needle-like structure increases significantly, which is accompanied by enhanced electric field line densities in the localized region (Fig. 1g, h). In contrast, the current density in all regions of $r$-Co$_3$S$_4$@NF with rod-like structure remains constant with sparse electric field line density around it. The essence of these phenomenon are electrostatic repulsion, where free electrons migrate to the sharpest regions on the charged electrodes, which increases the free electron density and further leads to a climb in the local electric field strength[17]. Kelvin probe atomic force microscopy experimentally confirmed that the electric field is higher for the $n$-Co$_3$S$_4$@NF with needle-like structure than the $r$-Co$_3$S$_4$@NF with rod-like structure (Supplementary Fig. 12). These results are consistent with the simulation results[17,28]. In addition, the S$^{2-}$ concentration on the surface of the $n$-Co$_3$S$_4$@NF electrode was further calculated using the Gouy-Chapman-Stern model, which shows that the ionic concentration on the electrode surface can be increased by a factor of about 1.9 when the voltage of the electrode material is increased from 400 to 1000 mV (Fig. 1i and Supplementary Fig. 13). We then experimentally evaluate the effect of tip structure in shaping the local environments. We performed the S$^{2-}$ absorbing test by measuring

the concentration of adsorbed S$^{2-}$ on the electrode (Supplementary Fig. 14). The ultraviolet and visible (UV-vis) absorption peak of S$^{2-}$ is located at the wavelength of 230 nm. The S$^{2-}$ absorption test shows that the number of S$^{2-}$ adsorbed by the $n$-Co$_3$S$_4$@NF electrode is more than twice that of the $r$-Co$_3$S$_4$@NF electrode. This absorbing test result is consistent with the simulation results. It indicates that the tip structure plays a crucial role in improving the attraction and mass transfer rate of ions in this region. CV curves at different scan rates (Supplementary Fig. 15) are measured to evaluate the electrochemically active surface area (ECSA). It is found that the $n$-Co$_3$S$_4$@NF electrode has a higher active area and better catalytic activity than the $r$-Co$_3$S$_4$@NF electrode (Supplementary Fig. 16)[26,29]. This indicates that the $n$-Co$_3$S$_4$@NF electrode has a larger electric-field-induced locally absorbed S$^{2-}$ concentration[17]. All tests predicted that the Co$_3$S$_4$ with a needle-like structure can exhibit excellent SOR activity.

**Electrocatalytic SOR performance**
The electrocatalytic SOR performance of as-prepared samples was studied in a typical H-type cell. The SOR catalytic activities of the $n$-Co$_3$S$_4$@NF electrode were first measured in 1 M NaOH with different Na$_2$S concentrations (0.1–1.5 M). During the electrolysis process, H$_2$ bubbles were continuously generated on the cathode, while the

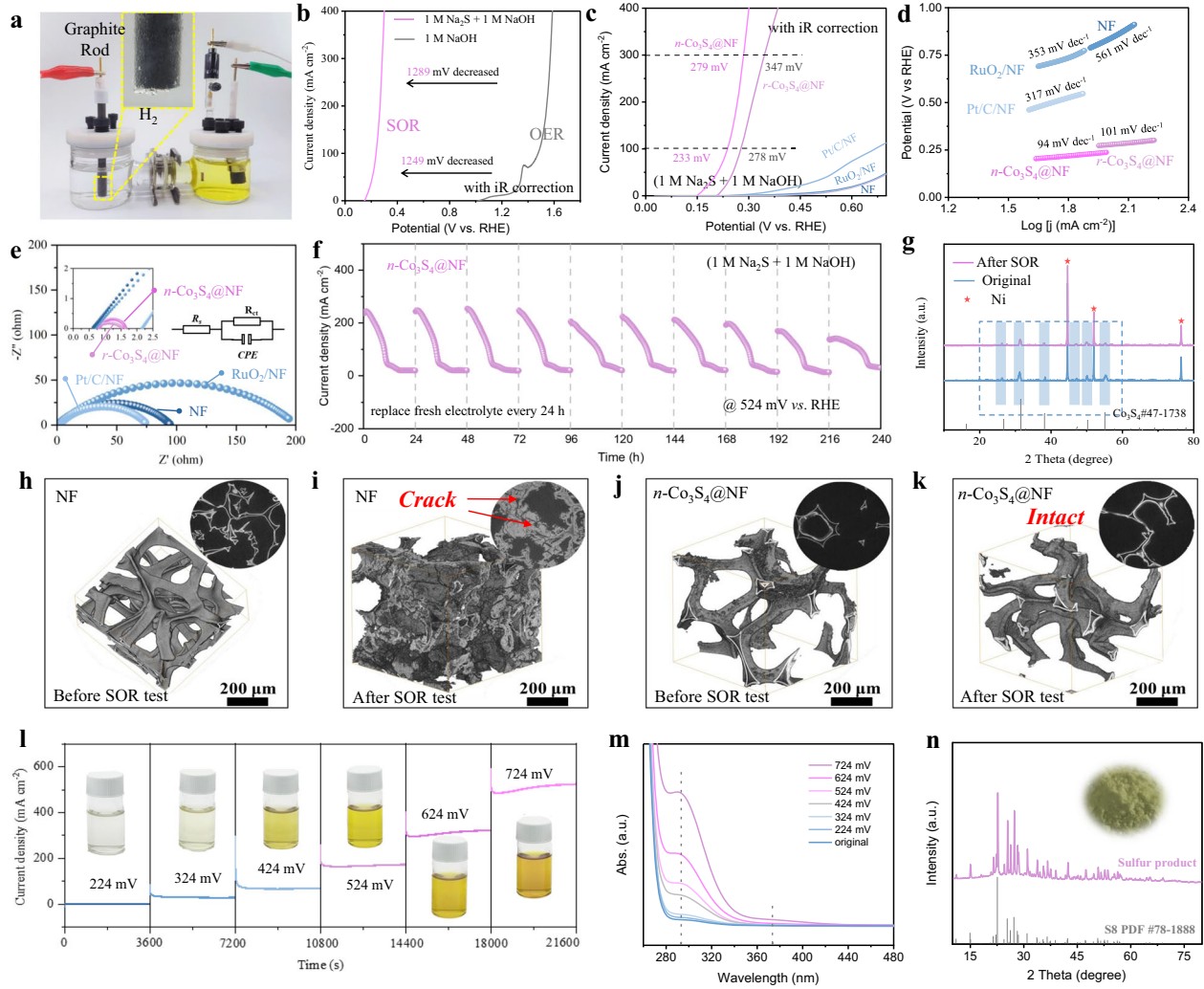

**Fig. 2 | *n*-Co$_3$S$_4$@NF SOR performance. a** The digital image of an H-type cell.
**b** Comparing the SOR and OER polarization curves (with *iR* compensation) of *n*-
Co$_3$S$_4$@NF with a loading mass of approximately 3.6 mg cm$^{-2}$ and a charge transfer
resistance (R$_{ct}$) of about 0.82 Ω. **c** LSV curves (with *iR* compensation), (**d**) Tafel
plots, and (**e**) Nyquist plots for SOR in 1 M NaOH containing 1 M Na$_2$S (pH ≈ 14.1).
**f** Durability test of *n*-Co$_3$S$_4$@NF for SOR (replacing fresh electrolyte every 24 h).
**g** XRD patterns of the *n*-Co$_3$S$_4$@NF before and after 240 h SOR durability test.

Three-dimensional X-ray tomography images of the electrodes: **h** before and (**i**)
after 24 h SOR stability test for blank NF; (**j**) before and (**k**) after 24 h SOR stability
test for *n*-Co$_3$S$_4$@NF. **l** Chronoamperometric responses of *n*-Co$_3$S$_4$@NF for SOR at
different potentials (inset: digital images of the electrolyte after electrolysis). **m** UV-
vis spectra of the electrolyte after 3600 s SOR of *n*-Co$_3$S$_4$@NF at different poten-
tials. **n** XRD pattern of the sulfur powder and the corresponding photograph.

electrolyte in the anode region gradually changed from colorless to
orange and gradually deepened over time (Fig. 2a). The average *n*-
Co$_3$S$_4$@NF loading on NF is around 3.6 mg cm$^{-2}$, and all the electrodes
size were around 1 cm$^{-2}$. The linear sweep voltammetry (LSV) curves
show the current density keeps increasing with increased S$^{2-}$ con-
centrations till 1 M (Supplementary Fig. 17). Therefore, the con-
centration of 1 M is selected for the SOR catalytic performance
measurement. Figure 2b illustrates that the *n*-Co$_3$S$_4$@NF required 1482
and 1568 mV (*vs.* RHE) to drive OER current densities of 100 and
300 mA cm$^{-2}$ in 1 M NaOH, respectively. It is worth noting that
although the OER catalytic activity of the *n*-Co$_3$S$_4$@NF exceeds that of
many previously reported electrocatalysts (Supplementary Table 1),
the required potential for SOR is significantly lower than that of the
anodic H$_2$O splitting (OER). As shown in Fig. 2c, the *n*-Co$_3$S$_4$@NF
electrode requires extremely low potentials of 233 and 279 mV (*vs.*
RHE) to afford SOR current densities of 100 and 300 mA cm$^{-2}$,
respectively. The SOR catalytic activity of *n*-Co$_3$S$_4$@NF is considerably
superior to that of other electrodes including *r*-Co$_3$S$_4$@NF, Pt/C/NF,
RuO$_2$/NF, and NF. Excitingly, it was found that our catalyst was able to

maintain a low potential (300 mV *vs.* RHE) even when higher current
densities (400 mA cm$^{-2}$) were required. There is a significant reduction
in the operating voltage of water splitting as a result of this advantage,
making it ideally suited for low-energy hydrogen production. In addi-
tion, *n*-Co$_3$S$_4$@NF exhibited the smallest Tafel slope of 94 mV dec$^{-1}$ and
the lowest charge transfer resistance of 0.82 Ω compared to other
catalysts in Fig. 2d, e and Supplementary Fig. 18. This reveals that the
unique structure of the needles can bring great advantages in terms of
kinetics. Long-term stability is a major challenge for the utility of SOR
catalysts in industrial production. We measured the stability of *n*-
Co$_3$S$_4$@NF by chronoamperometry at 524 mV for 240 h, in which the
electrolyte was changed every 24 h (Fig. 2f). It was observed that the
current density gradually decreased as the S$^{2-}$ concentration in the
solution decreased. However, the current density was maintained even
after ten times electrolyte changes, indicating its excellent stability for
SOR. As shown in Fig. 2g, the crystal structure of the *n*-Co$_3$S$_4$@NF
catalyst remained unchanged after a long-time electrocatalytic SOR
test. In addition, the electrode structures have also been investigated
by a three-dimensional X-ray tomography technique before and after a

24 h SOR durability test. It was observed that when driving the SOR with the conventional Ni foam, a commonly used electrode material for industrial electrolysis, its skeleton structure showed cracks and sulfur deposition due to the corrosive effect of $S^{2-}$ in the electrolyte and sulfur passivation, respectively (Fig. 2h, i). When driving the SOR with $n$-$Co_3S_4$@NF, its skeleton showed structural integrity after the SOR durability test (Fig. 2j, k). This suggests the long-term stability of the $n$-$Co_3S_4$@NF anode against $S^{2-}$ corrosion. In addition, the $n$-$Co_3S_4$@NF electrode shows no sulfur passivation due to its desulfurization effect. There were no significant changes in the crystal structure (Supplementary Fig. 19), morphology (Supplementary Fig. 20), and surface composition (Supplementary Figs. 21 and 22) of the $n$-$Co_3S_4$@NF electrode before and after the SOR durability test. In comparison to other SOR catalysts reported to date, the $n$-$Co_3S_4$@NF exhibits the pre-eminent SOR activity at the high current density (Supplementary Fig. 23 and Supplementary Table 2). Moreover, we performed chronoamperometric (CP) measurements for 1 h running at different potentials and used UV-vis spectrophotometry to analyze the products of corresponding electrocatalytic SOR (Fig. 2l, m). The measured current densities increased with the increase in applied potentials. All current-time curves exhibited negligible fluctuations further indicating the robust stability of the $n$-$Co_3S_4$@NF electrode during the electrocatalytic SOR process. The color of electrolytes is shown in Fig. 2l. With increasing electrolysis voltage, the original transparent electrolyte turns from light yellow to dark yellow, suggesting that the concentration of polysulfide intermediates gradually increases. The gradual rise of the absorption peak at 300 and 370 nm reflects the faster formation rate of short-chain polysulfide ions ($S_2^{2-}$-$S_4^{2-}$) in the electrolyte with increasing the applied potential (Fig. 2m)[30]. It is also clearly reflected in the color change of the electrolyte. In addition, the position of the absorption peak almost remains unchanged, suggesting that the electrolytic SOR product is unrelated to the applied potential in the testing range. Acidification of the dark yellow electrolyte after the SOR durability test resulted in bright yellow powdery products, of which the collected products (Fig. 2f) were confirmed to be monomeric sulfur through XRD (Fig. 2n). To determine the effectiveness of $n$-$Co_3S_4$@NF in treating realistic sulfion-containing sewage by electrocatalytic SOR method. The realistic sulfion-containing sewage was collected from a natural gas field located in Dazhou, Sichuan Province, PRC (Supplementary Fig. 24a). The realistic sulfion-containing sewage is directly used as the SOR electrolyte, in which the $n$-$Co_3S_4$@NF electrocatalyst requires a potential of 486 mV $vs$. RHE to achieve a current density of 100 mA cm$^{-2}$. The $n$-$Co_3S_4$@NF electrocatalyst can only achieve a current density of 150 mA cm$^{-2}$ at the potential of 500 mV $vs$. RHE. Due to strict environmental regulations and safety considerations, we can only collect diluted sulfur-containing wastewater from natural gas fields, resulting in a low concentration of sulfion, and the pH value of the solution and the concentration of sulfion are much lower than the simulated sulfion wastewater configured in the laboratory, therefore $n$-$Co_3S_4$ exhibits weak catalytic performance (Supplementary Fig. 24b, c).

## Electrocatalytic HER performance

The electrocatalytic HER performance of the as-prepared electrodes was evaluated in a typical three-electrode system. Figure 3a presented the LSV curves for various catalysts measured in 1 M NaOH electrolyte. Interestingly, even at current densities as high as 400 mA cm$^{-2}$, the overpotential of $n$-$Co_3S_4$@NF (311 mV), which is close to that of the state-of-the-art Pt/C/NF (295 mV) and superior to that of $r$-$Co_3S_4$@NF (347 mV). The intrinsic HER catalytic kinetics of the as-prepared catalysts were further revealed by the Tafel slope values. As shown in Fig. 3b and Supplementary Fig. 25a, the Tafel slopes for $n$-$Co_3S_4$@NF, $r$-$Co_3S_4$@NF, $n$-$Co(OH)_2$@NF, $r$-$Co(OH)_2$@NF, Pt/C/NF, and NF were 87, 107, 129, 146, 86, and 162 mV dec$^{-1}$ in 1 M NaOH, respectively. This

suggests that $n$-$Co_3S_4$@NF has favorable catalytic kinetics for the HER. The HER Tafel slope value of $n$-$Co_3S_4$@NF (87 mV dec$^{-1}$) suggests that the HER proceeds through a Volmer-Heyrovsky mechanism with the Heyrovsky step as the rate-limiting step[31]. On the other hand, $n$-$Co_3S_4$@NF highlighted the advantage of the lowest charge transfer resistance, suggesting that its needle-like structure promotes charge transfer kinetics and enhances the HER catalytic activity (Fig. 3c and Supplementary Fig. 26). Furthermore, the Faraday efficiency of hydrogen production of the $n$-$Co_3S_4$@NF electrode was evaluated using a drainage method. The experimental measurements showed $H_2$ yields in excellent agreement with the theoretical yields, achieving a high Faraday efficiency of 99.2% in 1 M NaOH and confirming the superior HER selectivity of the $n$-$Co_3S_4$@NF electrocatalyst (Fig. 3d). As shown in Fig. 3e, the $n$-$Co_3S_4$@NF electrocatalyst had excellent stability for HER. We also examined the crystal structure (Supplementary Fig. 19), morphology (Supplementary Fig. 20), and surface composition (Supplementary Figs. 21 and 22), of the catalyst after the durability test. It was found that the crystal structure of the $n$-$Co_3S_4$@NF catalyst was well preserved with no significant changes.

The corrosive nature of seawater owing to the presence of $Cl^-$ poses a greater challenge to the stability of electrocatalysts[32]. The seawater collected from Shenzhen Bay Park in Shenzhen PRC (Supplementary Fig. 27) was used to prepare the alkaline seawater electrolyte (1 M NaOH seawater). As shown in Fig. 3f, the zeta potential of $n$-$Co_3S_4$@NF (−24.5 mV) compared to $n$-$Co(OH)_2$@NF (30.4 mV) is shifted negatively. This suggests that the $n$-$Co_3S_4$@NF catalyst provides a weaker binding energy to $Cl^-$ and repels it[15]. This characteristic of the $n$-$Co_3S_4$@NF catalyst can alleviate its corrosion in seawater, making it a candidate for hydrogen production by seawater electrolysis. Tafel plots show the corrosion potential of $n$-$Co_3S_4$@NF is more positive than that of Ni foam in a 1 M NaOH seawater electrolyte (Supplementary Fig. 28), indicating a lower corrosion tendency and the improved corrosion resistance of the $n$-$Co_3S_4$@NF electrocatalyst compared to a blank Ni foam substrate[33]. The electrocatalytic HER performance of the prepared catalysts in an alkaline seawater electrolyte shows a decrease in activity compared to their performance in a 1 M NaOH electrolyte (Fig. 3g). This degradation can be attributed to the active sites' blockage and the degradation in conductivity of catalysts in the complex natural seawater environment. However, the $n$-$Co_3S_4$@NF still showed encouraging HER catalytic activity. Briefly, it required an overpotential of 262 mV to drive a current density of 100 mA cm$^{-2}$ with a Tafel slope of 132 mV dec$^{-1}$. This indicates that the $n$-$Co_3S_4$@NF demonstrates excellent HER activity and minimal degradation in a 1 M NaOH seawater electrolyte (Supplementary Fig. 25b). By testing the change of open circuit potential (OCP) along the CP test, the $n$-$Co_3S_4$@NF with high corrosion resistance shows negligible degradation over time (Supplementary Fig. 29). The durability of an electrocatalyst is a major challenge for the industrialization of seawater electrolysis. We assessed the durability of the $n$-$Co_3S_4$@NF catalyst in 1 M NaOH seawater. It showed remarkable long-term stability in alkaline seawater under 100 mA cm$^{-2}$ for 210 h, yielding only a 5.4% increase for the applied potential (Fig. 3i). Moreover, the Faraday efficiency of $n$-$Co_3S_4$@NF up to 99.1% reveals that no additional side reactions occur during hydrogen production in alkaline seawater, highlighting the high efficiency of the catalyst in HER. (Fig. 3h). Its backbone structure also retained substantial integrity upon the durability test, also indicating that the $n$-$Co_3S_4$@NF catalyst has excellent resistance to seawater corrosion (Supplementary Fig. 30). Simultaneously, we examined the crystal structures, morphology, and surface chemical states of the $n$-$Co_3S_4$@NF catalyst after the HER stability test in the alkaline seawater electrolyte (Supplementary Figs. 19–22). The $n$-$Co_3S_4$@NF catalyst showed negligible changes, presenting the vast potential of the $n$-$Co_3S_4$@NF catalyst for hydrogen production by seawater electrolysis.

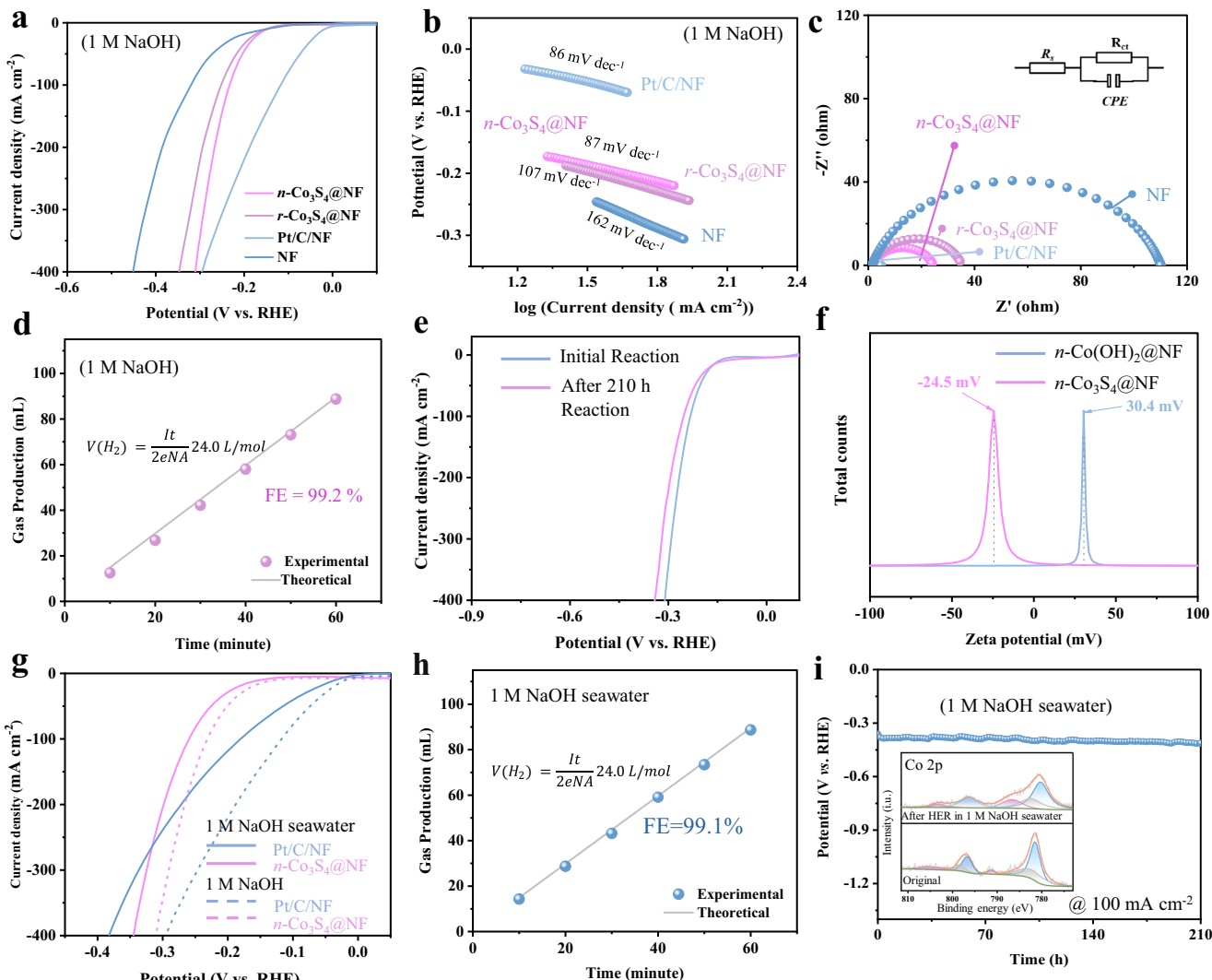

**Fig. 3 | n-Co₃S₄@NF HER performance. a** LSV curves (with *iR* compensation), (**b**) Tafel plots, (**c**) EIS plots for HER in 1 M NaOH. **d** Faradaic efficiency for hydrogen production of *n*-Co₃S₄@NF in 1 M NaOH (catalyst *n*-Co₃S₄@NF loading mass ≈ 3.6 mg cm⁻², R_ct ≈ 22.8 Ω). **e** LSV curves (with *iR* compensation) of the *n*-Co₃S₄@NF catalyst before and after the CP test in 1 M NaOH. **f** Zeta potentials of *n*-Co(OH)₂@NF and *n*-Co₃S₄@NF. **g** LSV curves (with *iR* compensation) for HER in 1 M NaOH and 1 M NaOH seawater (pH ≈ 14.2). **h** Faradaic efficiency of hydrogen production of *n*-Co₃S₄@NF in 1 M NaOH seawater. **i** HER durability test of *n*-Co₃S₄@NF under 100 mA cm⁻² in 1 M NaOH seawater (the inset shows the high-resolution XPS pattern of Co 2*p* before and after the durability test).

## Performance of hybrid seawater electrolyzer

Inspired by the remarkable catalytic performance of *n*-Co₃S₄@NF for SOR and HER, we used it as both cathode and anode in a SOR-assisted electrocatalytic hydrogen production system to obtain a typical hybrid seawater electrolyzer (Fig. 4a and Supplementary Fig. 31). When *n*-Co₃S₄@NF is used as both anode and cathode to assemble a two-electrode alkaline electrolyzer for overall freshwater and seawater splitting. The ASE electrolyzer can achieve the current density of 200 mA cm⁻² at cell voltages of 1566 and 1762 mV in 1 M NaOH and 1 M NaOH seawater, respectively (Fig. 4b, c). However, when *n*-Co₃S₄@NF is used as both anode and cathode to assemble a two-electrode alkaline electrolyzer for SOR-assisted overall freshwater and seawater splitting, the electrolyzer can achieve the current density of 200 mA cm⁻² at cell voltages of 551 and 575 mV, respectively. The driving voltage for the SOR-assisted electrocatalytic hydrogen production system was substantially lower compared to the conventional overall water and seawater splitting system. Moreover, compared with the ASE system (1575 mV), the needed electrolyzer voltage for the HSE system (506 mV) at the current density of 100 mA cm⁻² is reduced by 67.9% in 1 M NaOH seawater (Fig. 4c), highlighting its significant advantage in terms of low energy consumption. The long-term durability of the HSE hydrogen production system is shown in Fig. 4d. It is noticeable that the HSE remains stable for up to 504 h. This suggests the HSE system exhibits a great advantage of high efficiency and long-term stable operation. On the other hand, it is also far superior to state-of-the-art hydrogen production technologies such as alkaline water electrolysis, natural gas steam reforming, and methane steam reforming in terms of energy equivalent input and carbon dioxide equivalent emissions, underlining the sustainability, and high energy efficiency of HSE system for hydrogen production (Fig. 4e and Supplementary Table 3)[34,35]. Specifically, it requires only 1.21 kWh of electricity to produce per m³ H₂, representing a substantial energy advantage over currently reported water-splitting electrolyzers for hydrogen production, which couples the HER and various alternative anodic reactions (Fig. 4f and Supplementary Table 4). In addition, the unique needle structure of the *n*-Co₃S₄@NF catalyst endows the HSE system with multiple advantages such as high mass transfer efficiency, low operating voltage, and long-term durability, which will allow for robust and sustained hydrogen production. Simultaneously, the coupled SOR in the HSE system facilitates the recycling of sulfion-

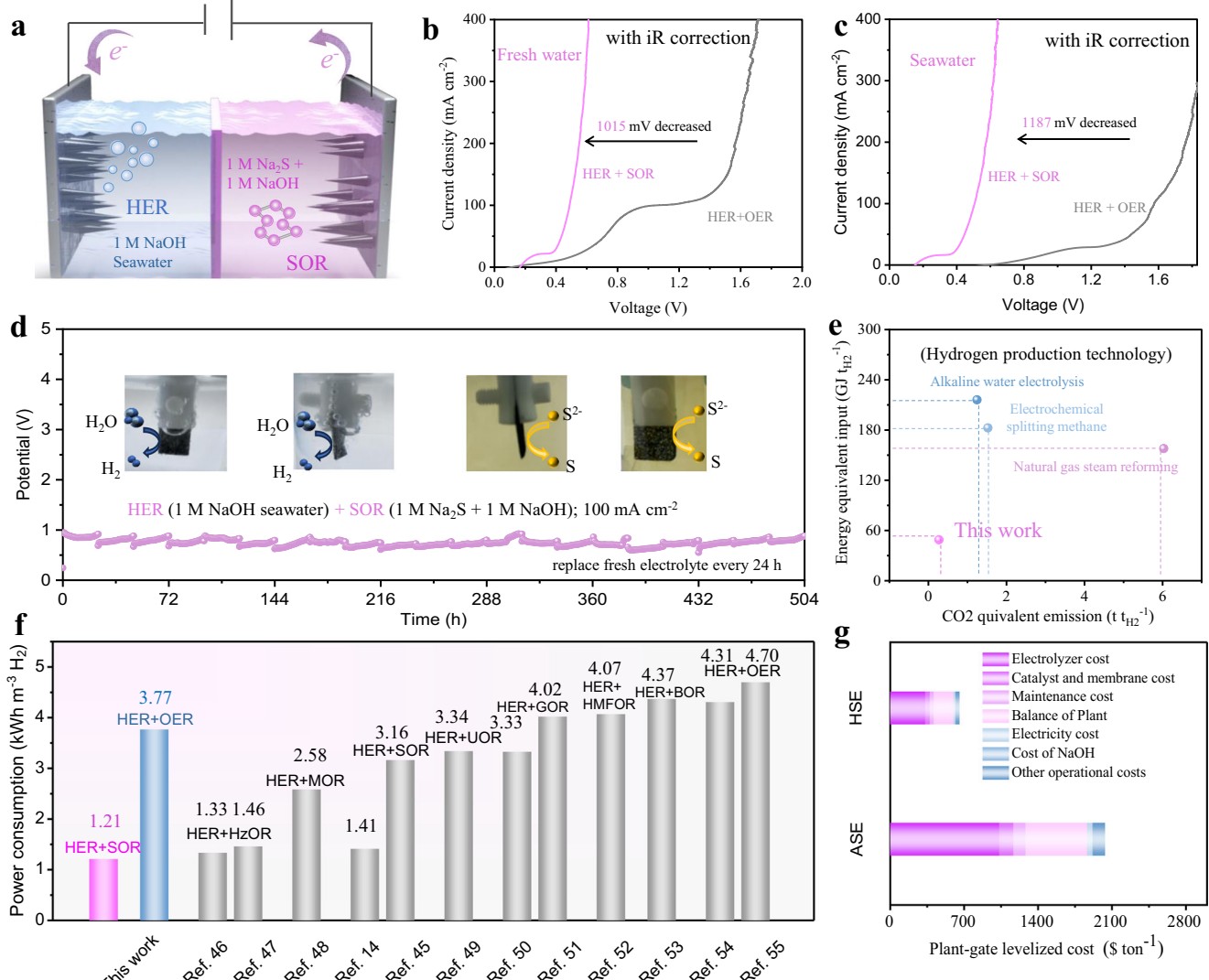

**Fig. 4 | HSE electrochemical performance and techno-economic analysis.**
**a** Schematic diagram of the HSE system using bifunctional *n*-Co₃S₄@NF (loading mass ≈ 3.6 mg cm⁻²) electrode in 1 M NaOH seawater for HER (left) and in 1 M Na₂S + 1 M NaOH (pH ≈ 14.1) for SOR (right). LSV curves for ASE and HSE in (**b**) 1 M NaOH (pH ≈ 14.0, with *iR* compensation) and (**c**) 1 M NaOH seawater (pH ≈ 14.2, with *iR* compensation) using *n*-Co₃S₄@NF as both the anode and cathode. The R꜀ₜ for the HER coupling SOR in a pure water system is approximately 18.0 Ω, while the R꜀ₜ for the HER coupling OER in the same system is around 20.8 Ω. The R꜀ₜ for ASE is about

20.8 Ω, and for HSE it is about 19.0 Ω. **d** Durability test of the HSE system assembled by bifunctional *n*-Co₃S₄@NF under 100 mA cm⁻² (replacing fresh electrolyte every 24 h). **e** A comparison of the HSE system with different hydrogen production techniques in energy equivalent input and CO₂ equivalent emission. **f** Comparison of the HSE with reported water-splitting electrolyzers coupling the HER and various anodic reactions[14,45–55]. **g** A rough techno-economic analysis of the HSE and ASE systems for hydrogen production.

containing sewage to obtain high-value elemental sulfur, which is of great significance for environmental protection and increasing the high-value output of this system. A rough techno-economic analysis shows that the HSE system can significantly reduce the total cost per ton of hydrogen production by up to 67.7% compared to the traditional ASE system, indicating its promising application prospects (Fig. 4g and Supplementary Table 5)[36]. Therefore, it will contribute to the realization of the goal of "Carbon Neutrality and Carbon Peaking".

**Catalyst application exhibition in HSE**
A cost-effective, environment-friendly, and sustainable HSE system for hydrogen production might be scaled up by feeding toxic sulfion-containing sewage and valueless seawater into the HSE device in the industrial region with abundant sulfion-rich sewage (Fig. 5a). To demonstrate the practicality of the HSE system for sustainable hydrogen production, we used a thermoelectric generator (TEG), which can use the waste heat from industrial production activities to

generate electricity, to power it (Fig. 5b). We found that the TEG-driven HSE system was able to maintain stable voltage and current density 1 M NaOH and in natural alkaline seawater up to 210 min, showing its long-term practicality (Supplementary Fig. 34a and Fig. 5c). On the other hand, integrating the HSE system to a solar cell showed even better stability (Fig. 5d and Supplementary Movie 1), which allows long-term stable operation at an average voltage of 890 mV (current density of 34 mA cm⁻²) in natural alkaline seawater (Fig. 5e). On this basis, by using seawater and industrial sulfion-containing wastewater into renewable energy power HSE in coastal areas with strong solar radiation or strong wind patterns, the burden of hydrogen production on electrical grids can be removed and sustainable hydrogen production scale can be expanded. H₂S molecule is an abundant resource but is usually considered a toxic byproduct in chemical feedstocks such as natural gas, syngas, and refinery gas, is a promising hydrogen source[37]. The electrocatalytic decomposition of H₂S is a mild and efficient method, which involves the anodic oxidation of S²⁻ and the

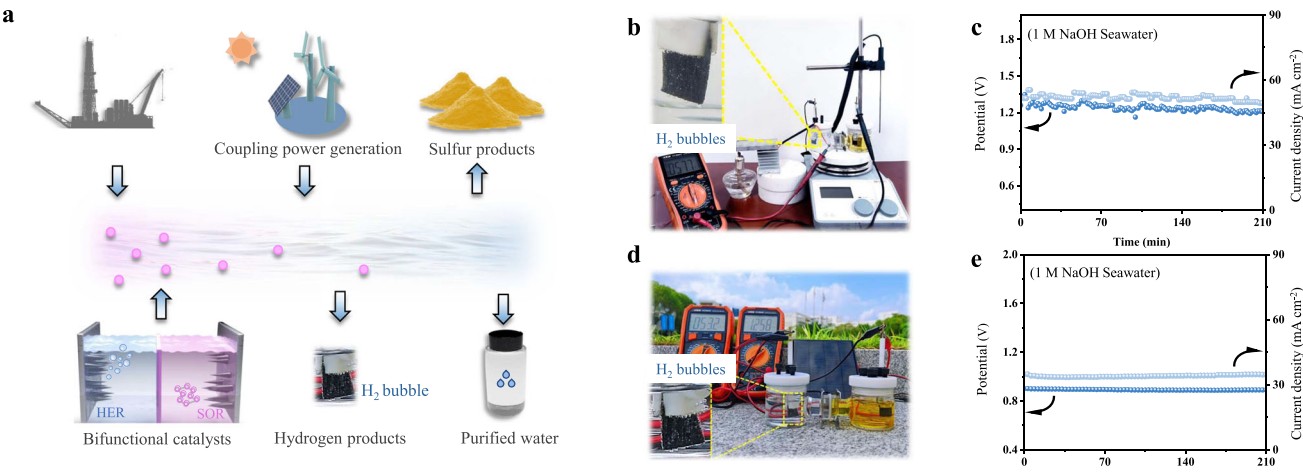

**Fig. 5 | Catalyst's applications in HSE. a** Schematic illustration of hydrogen production by renewables-powered HSE with costless seawater and industrial sulfion-containing sewage as the feeds. **b** The digital image of a TEG-driven HSE. **c** The current density or voltage versus time curves of this TEG-driven HSE in 1 M NaOH seawater. **d** The digital image of a solar cell driven HSE. **e** The current density or voltage versus time curves of this solar cell driven HSE in 1 M NaOH seawater.

simultaneous cathodic evolution of $H_2$. Due to the realistic sulfion-containing wastewater we collected from the natural gas field has undergone dilution and is not suitable for direct use as an electrolyte for SOR (Supplementary Fig. 24). To determine the effectiveness of $n$-$Co_3S_4$@NF in SOR-assisted seawater electrolysis hydrogen production and sulfion-containing sewage purification in a realistic system, we constructed a demo HSE, in which $n$-$Co_3S_4$@NF is used as both anode and cathode, for the electrocatalytic selective removal of $H_2S$ from simulated industrial syngas (49% CO, 49% $H_2$, and 2% $H_2S$)[38]. As shown in Supplementary Fig. 35, UV-vis spectra show that the sulfion's concentration of 2% $H_2S$/syngas + 1 M NaOH is comparable to that of the simulated sulfion-containing wastewater (1 M $Na_2S$ + 1 M NaOH). The LSV test showed that the HSE using 2% $H_2S$/syngas + 1 M NaOH as anolyte had a high response current density, which is comparable to that of the HSE using 1 M $Na_2S$ + 1 M NaOH as anolyte (Supplementary Fig. 36). Furthermore, the long-term galvanostatic test at 100 mA cm$^{-2}$ current density showed that the HSE using 2% $H_2S$/syngas + 1 M NaOH as anolyte could be well maintained for over 120 h (Supplementary Fig. 37). This suggests its excellent stability in the selective removal of $H_2S$ from industrial syngas and energy-saving hydrogen production.

## Discussion

In summary, we successfully fabricated a bifunctional $n$-$Co_3S_4$@NF catalyst with needle-like structures via a simple two-step hydrothermal method for SOR-assisted energy-saving hydrogen production by seawater electrolysis. Theoretical calculations and experimental validation showed that the unique tip effect could greatly enhance the current density in the tip region, which effectively boosted the mass transfer rate in the SOR process. The HSE system assembled by the bifunctional $n$-$Co_3S_4$@NF is not only able to reduce power consumption up to 67.9% compared to the conventional ASE system but can also be grid-connected to a thermoelectric/photovoltaic power generation system for efficient and long-lasting hydrogen production. This work provides a new idea for the design of efficient catalysts, an economical way for energy conversion from waste chemical energy to valuable hydrogen, and simultaneously achieving sulfion-rich wastewater purification and sulfur recovery.

## Methods
### Materials and chemicals
Sodium hydroxide (NaOH, 95%), cobalt (II) nitrate hexahydrate ($Co(NO_3)_2 \cdot 6H_2O$, 99%), urea ($CO(NH_2)_2$, 99.5%), ammonium fluoride

($NH_4F$, 99.99%), and sodium sulfide nonahydrate ($Na_2S \cdot 9H_2O$) were acquired from Macklin. Hydrochloric acid (HCl), methenamine ($C_6H_{12}N_4$, 99%), sulfuric acid ($H_2SO_4$), hexamethylenetetramine (HMT) and ethanol were acquired from Aladdin. The Nafion solution (5%), $RuO_2$ (with a Ru content >75%), and Pt/C (20 wt%) were purchased from Suzhou Sinero Technology Co., LTD. The nickel foam (NF, thickness: 1.0 mm; aperture: 0.1 mm; porosity: 97.2%) used in this work has a thickness of 1.0 mm and is obtained from Shanghai Tankii Alloy Material Co., Ltd. NF has been cleaned sequentially with 1.0 M HCl, ethanol, and deionized water (DI water) before being used. DI water was used in all experimental sections, having a resistance of 18.25 MΩ cm.

### Synthesis of $n$-$Co(OH)_2$@NF precursor
In a Teflon-lined stainless autoclave with a capacity of 100 mL, 5.0 mmol of $Co(NO_3)_2 \cdot 6H_2O$, 30.0 mmol of $CO(NH_2)_2$, and 12.5 mmol of $NH_4F$ were dissolved in 50.0 mL of DI water. Subsequently, a thoroughly cleaned and dried piece of NF (10 × 50 mm², thickness: 1.0 mm; aperture: 0.1 mm; porosity: 97.2%) was immersed in the solution. The hydrothermal reaction was maintained at 120 °C for 6 h. Then, the obtained $n$-$Co(OH)_2$@NF precursor was washed with DI water and ethanol.

### Synthesis of $n$-$Co_3S_4$@NF catalyst
In a Teflon-lined stainless autoclave with a capacity of 100 mL, 1 g of $Na_2S \cdot 9H_2O$ was dissolved in 40 mL of DI water. Subsequently, the $n$-$Co(OH)_2$@NF precursor (10 × 10 mm², thickness: 1.0 mm; aperture: 0.1 mm; porosity: 97.2%) was immersed in the solution. The Teflon-lined stainless autoclave was maintained at 90 °C for 12 h. Then, the obtained $n$-$Co_3S_4$@NF was washed with DI water and ethanol. The loading mass of the synthesized $n$-$Co_3S_4$@NF was approximately 3.6 mg cm$^{-2}$.

### Synthesis of $r$-$Co(OH)_2$@NF precursor
In a Teflon-lined stainless autoclave with a capacity of 100 mL, 1.45 g of $Co(NO_3)_2 \cdot 6H_2O$, 1.5 g of HMT, and 0.37 g of $NH_4F$ were dissolved in 50 mL of DI water. Subsequently, a thoroughly cleaned and dried piece of NF (10 × 50 mm², thickness: 1.0 mm; aperture: 0.1 mm; porosity: 97.2%) was immersed in the solution. The Teflon-lined stainless autoclave was maintained at 95 °C for 24 h. Upon completion of the reaction, the resulting $r$-$Co(OH)_2$@NF precursor was washed with DI water and ethanol.

## Synthesis of r-Co₃S₄@NF catalyst

Consistent with the n-Co(OH)₂@NF precursor ($10 \times 10$ mm², thickness: 1.0 mm; aperture: 0.1 mm; porosity: 97.2%) sulfation conditions, the r-Co(OH)₂@NF precursor was sulfated. The loading mass of r-Co₃S₄@NF was approximately 4.5 mg cm⁻².

## Synthesis of 20 wt% Pt/C/NF

20 mg of 20 wt% Pt/C powder, 500 μL of ethanol, and 50 μL of Nafion were sonicated for 30 min. Next, 100 μL of the above dispersion solution was slowly dropped on cleaned NF ($10 \times 10$ mm², thickness: 1.0 mm; aperture: 0.1 mm; porosity: 97.2%) and dried in air for 10 h to prepare the Pt/C/NF electrode.

## Synthesis of RuO₂/NF

20 mg of 20 wt% RuO₂ powder, 500 μL of ethanol, and 50 μL of Nafion were sonicated for 30 minutes. Next, 100 μL of the above dispersion solution was slowly dropped on cleaned NF ($10 \times 10$ mm², thickness: 1.0 mm; aperture: 0.1 mm; porosity: 97.2%) and dried in air for 10 h to prepare the RuO₂/NF electrode.

## Material characterizations

Scanning electron microscope (SEM) images were obtained by HITACHI SU8010. TEM images were characterized by FEI TalosF200S G2 at 200 kV. XRD patterns were obtained by Bruker D8 Advance at a rate of 2° min⁻¹. XPS patterns were performed on an Escalab Xi + system. Raman spectra were performed on a Renishaw inVia. UV-vis spectral analysis maps were obtained from a Cary 5000. Zeta potentials were performed under a Brookhaven 90Plus PALS device. The X-ray tomography tests were performed on an Xradia Versa XRM-500 system[39]. 1600 projections were recorded while the sample was rotated about 360 degree. The pixel size of the reconstructed 3D image is about 1.09 μm. The 3D data was processed and visualized using Avizo Fire software.

## Electrochemical test

All electrochemical experiments were performed on an electrochemical workstation (CHI 660E, CH Instruments, Inc.). The electrode size used in testing is 1 cm⁻². All mentioned potentials of SOR, HER and OER were converted to potentials relative to reversible hydrogen electrode (RHE) according to E vs. RHE = E vs. Hg/HgO + 0.059 × $pH$ + 0.098 V. The electrolytes involved in this work including 1 M NaOH, 1 M Na₂S + 1 M NaOH, and 1 M NaOH seawater (seawater was collected from Shenzhen Bay Park (Shenzhen, China) and centrifuged several times to remove visible impurities before use). Linear sweep voltammetry (LSV) curves were measured at a scan rate of 5 mV s⁻¹ (with iR compensation, unless specifically emphasized). The electrochemical SOR and HER dynamics of these catalyst is investigated through the Tafel slope derived from the corresponding LSV curves. The resistance was determined by constant potential electrochemical impedance spectroscopy (EIS) at an amplitude of 5 mV over a frequency range of 10⁶ to 0.1 Hz. The CV was tested with different scan rates (20, 30, 40, 50 and 60 mV s⁻¹) from 100 to 200 mV (vs. RHE) to calculate the effective ECSA of the catalyst. A linear plot was obtained by plotting the value of $\Delta j$ ($|j_{max} - j_{min}|/2$) of the current density against the CV scan rate. The slope of the fitted line is $C_{dl}$, which is proportional to the electrochemical surface area.

## Half-cell SOR test

The experimental setup employed a Hg/HgO electrode as the reference electrode, a graphite rod (Gaossunion) as the counter electrode, and the prepared sample as the working electrode. The anode electrolyte consists of 1 M Na₂S + 1 M NaOH, with a pH around 14.1, while the cathode electrolyte is 1 M NaOH, with a pH around 14. The two electrolytes are divided by a Nafion 117 (N-117) membrane. Both chambers had a solution volume of 35 mL (total chamber volume:

50 mL). The performance of all catalysts was evaluated by linear sweep voltammetry (LSV) measurements at a scan rate of 5 mV s⁻¹ with iR-compensation. Stability tests were carried out by using the Amperometric i-t Curve (i-t) program. The potential was set at 524 mV (vs. RHE, without iR compensation), and the electrolyte was changed every 24 h (without iR compensation). EIS tests were performed at a potential of 300 mV (vs. RHE).

## UV-Vis detection

After each chronoamperometric test, the resulting solution on the SOR side was collected (after standing at room temperature for 20 min) and for each UV-Vis detection, 10 μL of the electrolyte solution containing products was diluted to 1.0 mL with DI water.

## Harvest of sulfur product

After the SOR, H₂SO₄ was added dropwise to the electrolyte in the presence of an ice bath until the pH of the electrolyte was 1. The pH was tested by a pH meter (INESA Scientific Instruments Co., Ltd., Shanghai, China) at 25 °C. After centrifugation at 10,000 rpm for 30 min, the yellow precipitate was taken and the precipitate was placed in a vacuum oven at 60 °C for 24 h.

## Half-cell HER test

The experimental setup employed a Hg/HgO electrode as the reference electrode, a graphite rod (Gaossunion) as the counter electrode, and the prepared sample as the working electrode. The electrolyte was 1 M NaOH or 1 M NaOH seawater. EIS tests were performed at a potential of 150 mV (vs. RHE). The LSV measurements at a scan rate of 5 mV s⁻¹ with iR-compensation. Stability tests were obtained by chronoamperometry (CP) with guaranteed current density at 100 mA cm⁻².

## Full cell (alkaline-alkaline SOR/HER cell, alkaline-alkaline OER/HER cell) test

The experiments were performed using an H-type electrolytic cell, and the volume of both cation and anion side solutions was set to 35 mL (cathode volume: 50 mL). The alkali-alkali SOR/HER cell used a two-electrode system with 1 M Na₂S + 1 M NaOH as the anode side electrolyte and 1 M NaOH seawater or 1 M NaOH as the cathode side electrolyte, both the two electrodes are n-Co₃S₄@NF. The alkaline-alkaline OER/HER cell uses the same device and electrolyte volume. The LSV measurements at a scan rate of 5 mV s⁻¹ with iR-compensation. The alkali-alkali SOR/HER stability test was obtained by CP procedure ensuring current density at 100 mA cm⁻² and changing the electrolyte on both sides every 24 h. The alkali-base OER/HER stability test at a current density of 100 mA cm⁻² was tested by the CP program.

## Faradaic efficiency measurement

Faradaic efficiency (FE) was measured in a sealed two-electrode H-type cell with a total of two FE tests, the first test HER-side and OER-side electrolytes were 1 M NaOH, the second FE test electrolyte cathode side electrolyte was 1 M NaOH seawater and anode side electrolyte was 1 M Na₂S + 1 M NaOH. n-Co₃S₄@NF catalyst was used as cathode and anode. Timing tests were performed at a current of 200 mA to continuously produce hydrogen and oxygen. The generated H₂ was collected using the drainage method in a measuring cylinder filled with water. To evaluate the FE, the theoretical H₂ produced versus time was:

$$H_2 Production = \frac{It}{2eN_A} \times 24.0 \, \text{L/mol} \qquad (1)$$

where $I$ is the current, t is time, e is the meta-charge, and $N_A$ is Avogadro's constant. The temperature in the laboratory was about 20 °C, so the molar volume of the gas was determined to be 24.0 L mol⁻¹. FE

was calculated using the following equation:

$$FE = \frac{Experimental\ gas\ yield}{Theoretical\ gas\ yield} \times 100\% \quad (2)$$

## $S^{2-}$ adsorption measurement

The concentrations of adsorbed $S^{2-}$ on electrodes were performed in 1 M NaOH containing 1 M $Na_2S$ solution by using a three-electrode system and UV-vis measurements[40]. All the electrodes were run in 1 M NaOH containing 1 M $Na_2S$ solution with an applied voltage of 0.4 V *vs.* RHE. When the running time reached 120 s, the electrode was directly raised above the electrolyte and then transferred with voltage and immersed in 10 mL of pure water. Next, the applied potential was removed, and shaking the electrode lasted for 1 min in pure water, to enable the adsorbed $S^{2-}$ on the surface of catalysts to be completely released into the pure water. The concentration of $S^{2-}$ in the water was checked using UV-vis measurements after repeating the above process.

## Kelvin probe force microscopy (KPFM) measurement

KPFM measurement is a powerful technique for simultaneous obtaining the topography and surface potential with nanometer scale spatial resolution and millivolt sensitivity in potential resolution via a dual pass process. The KPFM measurement was implemented based on an AFM (NT-MDT NTEGRA) system using a conductive NSG30/Pt tip. All the images were recorded in semi contact mode at room temperature with a relative humidity of 30%. Si cantilevers (NT-MDT with a typical curvature radius of the tip of 35 nm and a typical resonant frequency of 0.5 kHz) were used. In a standard KPFM procedure, a DC bias voltage is continually adjusted with a potential feedback loop to nullify the surface potential difference between the probe and sample. During the SemiContact 2-pass KPFM measurement, an AC bias voltage, superimposed on a DC bias voltage, is applied between the probe and sample. Thus, the potential difference can be written as

$$V = V_{dc} - \Delta V + V_{ac}\sin(\omega t) \quad (3)$$

where $V_{dc}$ and $V_{ac}$ are DC bias and AC bias, respectively. The $\Delta V = (\phi_{sample} - \phi_{tip})/e$ is the contact potential difference (CPD) between the probe and sample.

## Finite element analysis (FEA)

We used the COMSOL Multiphysics solver to simulate the current density distribution and the $S^{2-}$ concentration distribution under the preset electric field in the electrolyte around the surface of different catalysts, which is simplified in the conical shape to investigate the local situations[17]. The dimensions of these systems are set at the scale of ten nanometers and one micrometer.

The current density distribution is simulated by the steady state research in the 'primary current distribution' module. The electrolyte conductivity was assumed to be 5 S $m^{-1}$. The potential losses due to electrode kinetics and mass transport are assumed to be negligible, and ohmic losses are governed by the current distribution in the cell. The current density was computed using the equation:

$$\nabla i_l = Q_l, i_l = -\sigma_l \nabla \Phi l \quad (4)$$

where $i_l$ represents the local current density in the electrolyte, $\sigma_l$ represents the assumed electrolyte conductivity, and $\Phi_l$ represents the preset voltage of 50 mV.

The concentration distribution under the electric field migration around the catalyst surface is simulated by the transient state research in 'dilute material transfer' coupled with a 'primary current

distribution' module, which was conducted to provide the preset electric field with the given voltage of 1.0 V. The $S^{2-}$ concentration distribution was simulated by the equation[16,41]:

$$\frac{\partial c_i}{\partial t} + \nabla J_i = R_i J_i = -D_i \nabla c_i - z_i u_{m,i} F c_i \nabla V \quad (5)$$

where $\nabla \cdot J_i$ represents the contribution from the external factors, including the concentration gradient and ion migration in the electric field in this model. $D_i$ represents the diffusion coefficient, which is set as $10^{-9}$ $m^2$ $s^{-1}$ here. $z_i$ is the ion charge of $S^{2-}$, and $u_{m,I}$ represents the ion mobility of $10^{-13}$ s mol $kg^{-1}$.

## Theoretical simulation

The first-principles are employed to perform density functional theory (DFT) calculations within the generalized gradient approximation (GGA) using the Perdew–Burke–Ernzerhof (PBE) formulation, as implemented in the CASTEP package[42–44]. The convergence tolerance for geometry optimizations was $2.0 \times 10^{-5}$ eV/atom, 0.05 eV/Å, and 0.002 Å for the energy change, maximum force, and maximum displacement, respectively. The vacuum spacing in a direction perpendicular to the plane of the structure is 20 Å for the surface. Finally, the adsorption energies ($E_{ads}$) are calculated from the following equation.

$$E_{ads} = E_{ad/sub} - E_{ad} - E_{sub} \quad (6)$$

where $E_{ad/sub}$, $E_{ad}$, and $E_{sub}$ are the total energies of the optimized adsorbate/substrate system, the adsorbate in the structure and the clean substrate, respectively.

The free energy is calculated from the following equation.

$$G = E_{adS} - ZPE - TS \quad (7)$$

where G, $E_{ads}$, ZPE and TS are the free energy, total energy from DFT calculations, zero point energy and entropic contributions, respectively.

## Data availability

All data that support the findings of this study are presented in the Manuscript and Supplementary Information, or are available from the corresponding author upon reasonable request. Source data are provided with this paper.

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

## Acknowledgements

G.Z. appreciates the support from the Joint Funds of the National Natural Science Foundation of China (U21A20174), Shenzhen Science and Technology Program (KQTD20210811090112002), Guangdong Innovative and Entrepreneurial Research Team Program (2021ZT09L197), Guangdong Basic and Applied Basic Research Foundation (2023B1515120099), Interdisciplinary Research and Innovation Fund of Tsinghua Shenzhen International Graduate School, and the Tsinghua Shenzhen International Graduate School-Shenzhen Pengrui Young Faculty Program of Shenzhen Pengrui Foundation (SZPR2023007).

## Author contributions
G.Z., T.L., and B.W. conceived the idea and designed the project. G.Z., T.L., B.W., Y.C., Z.L., S.W., Q.Z., and J.S. supervised the experiments and edited the paper. T.L., B.W., and S.W. performed the catalyst synthesis and tested the catalysts. Y.C. contributed to the density functional theory calculations parts. Z.L. conducted the finite element analysis simulations. All authors analyzed the data and discussed the results.

## Competing interests
The authors declare no competing interests.
