## [Peer Review File · Nature Communications]

REVIEWER COMMENTS

Reviewer #1 (Remarks to the Author):

The authors reported the coupling of hydrogen evolution reaction and the sulfion oxidation reaction toward hydrogen production from seawater. Needle-like Co₃S₄ catalyst grown on nickel foam was employed for the SOR and the HER. The local field enhancement at the tip of the needle-like structure was regarded as the major reason for the enhanced electrocatalytic performance. I suggest it could be rejected due to the lack of novelty and the inadequate evidence.

1. More experimental evidences for the enhanced local electric field should be provided, as it is the primary argument of this work. The finite element numerical calculation alone is not adequate. Besides, details for the finite element numerical calculation should be provided in the experimental section.

2. The citations need improvement. More relevant reference should be cited at proper place such as line 100, line 150, line 155, line 159, line 277... Also, are there any reference in which the local electric field and the ion concentration are analyzed by the finite element numerical calculation? If yes, relevant reference should be cited.

3. The analysis on the S 2p XPS spectra is not reasonable (line 138-141). In figure S9b and S10b, the peak at around 169 eV indicates the presence of SO₄²⁻ species. Also, it is not necessary to include Co-S peaks in the S 2p spectra. They are generally included in the S 2p_{1/2} and S 2p_{3/2} peaks.

4. In the Co 2p XPS spectra, why the peak shift of n-Co₃S₄@NF is more obvious than r-Co₃S₄@NF?

5. The adsorption of reactants such as OH⁻, S₂⁻, Cl⁻ on catalyst surface is not solely determined by the electrostatic interaction. The zeta potential values cannot be used as the indicator for the interaction between Cl⁻ and catalyst (line 276).

6. The sulfion concentration (1M) used in the electrolyzer is too high. What is the sulfion concentration of typical sulfion sewage? The authors are encouraged to collect industrial sulfion sewage and used in the electrolyzer so as to provide a more practical demonstration.

Reviewer #2 (Remarks to the Author):

This manuscript fabricated a bifunctional n-Co₃S₄@NF catalyst with a needle-like structure for SOR-assisted energy-saving hydrogen production by seawater electrolysis. The HSE system assembled by the bifunctional n-Co₃S₄@NF is not only able to reduce power consumption up to 67.9% compared to the conventional ASE system but can also be grid-connected to a thermoelectric/photovoltaic power generation system for efficient and long-lasting hydrogen production. The results from multiple characterizations and calculations are reliable. Besides, the plots and figures are well-designed and shown in a logical manner. Overall, this work is interesting and this manuscript is well organized, I would be pleased to recommend that it be published after minor revision. Detailed comments are listed as follows:

1. Regarding the DFT calculation, structural models of reaction intermediates adsorbed on the catalyst should be provided.

2. The details for Comsol simulations should be provided.

3. The authors have conducted sufficient theoretical and experimental research on the promotion of SOR

performance by needle-like structure design of catalyst. Is the needle-like structure design of the catalyst beneficial for HER performance? Please add more discussions about it.

4. The annotation text and scale in the figure should be clear. For example, Fig. 3d, Fig S19b, and Fig. 23.

5. The authors are suggested to explain why there appears fluctuations in durability test of the HSE and ASE systems in Figure 4d;

6. As SOR can proceed in different way, one may be curious why electrochemical reaction ($S^{2-} - 2e^- \rightarrow S$) or products (S) are dominated in the present work, and how the authors regulated no further oxidation happens on S.

Reviewer #3 (Remarks to the Author):

In this manuscript, the authors design a needle-like Co₃S₄ bifunctional catalyst grown on nickel foam (NF) with a unique tip structure (n-Co₃S₄@NF), which can achieve energy-saving hydrogen production, sulfion-rich wastewater purification, and sulfur recovery. Both experiments and calculations have been applied to explain the performance. This study is novel and very interesting. The data and methodology are reliable and the manuscript is also well organized and presented clearly. The conclusion is also clear and reliable. This manuscript can be considered for publication after addressing the following minor issues.

1. The HSE system rate by the 1.0 V commercial solar cell should provide a video.

2. There are some typos in the manuscript, which should be corrected. For example, "but is far higher than that of n-Co(OH)₂@NF and r-Co(OH)₂@NF"; in Figure 3e and Figure 3i, "vs. RHE".

3. Figure 4g is slightly blurry and should be improved appropriately.

4. The electro-oxidation products (elemental sulfur) of the sulfur oxidation reaction are prone to passivating the electrode surface and increasing the reaction overpotential, even making continuous operation infeasible. Can the design of the needle structure weaken this passivation effect?

Response to the Reviewer #1

General Comments:

The authors reported the coupling of hydrogen evolution reaction and the sulfion oxidation reaction toward hydrogen production from seawater. Needle-like Co_3S_4 catalyst grown on nickel foam was employed for the SOR and the HER. The local field enhancement at the tip of the needle-like structure was regarded as the major reason for the enhanced electrocatalytic performance. I suggest it could be rejected due to the lack of novelty and the inadequate evidence.

Response. Thanks for your helpful comments on our manuscript. Concerning your important concerns and advice, we have added experiments and characterizations to address them as much as possible. After careful checking and consideration, we have made substantial revisions to the revised manuscript (MS) and supplementary information (SI). Now we believe that the main results of this study are more clear and reliable. Below are the point-to-point responses to your comments.

Original comment 1-1:

More experimental evidences for the enhanced local electric field should be provided, as it is the primary argument of this work. The finite element numerical calculation alone is not adequate. Besides, details for the finite element numerical calculation should be provided in the experimental section.

Response 1-1. Thank you for your suggestion. We have provided more experimental evidence for the enhanced local electric field, including S^{2-} adsorption tests and Kelvin probe force microscopy (KPFM) tests. These tests and finite element simulations corroborate the contributions from a strong electric field at the tip. Besides, details for the finite element numerical calculation have been provided in the experimental section of the revised SI.

Specifically, we experimentally evaluate the effect of tip structure in shaping the local environments. We performed the S^{2-} absorbing test by measuring the

concentration of adsorbed S^{2-} on the electrodes (**Fig. R1**). The UV-vis absorption peak of S^{2-} is located at the wavelength of 230 nm. The S^{2-} absorption test shows that the number of S^{2-} adsorbed by the $n\text{-Co}_3\text{S}_4@\text{NF}$ electrode is more than twice that of the $r\text{-Co}_3\text{S}_4@\text{NF}$ electrode. The electrochemically active surface area (ECSA) was found to have a positive relationship with the double layer capacity (C_{dl}), but the C_{dl} value of the $n\text{-Co}_3\text{S}_4@\text{NF}$ electrode is less than twice that of the $r\text{-Co}_3\text{S}_4@\text{NF}$ electrode (**Figs. R2 and R3**). This indicates that the $n\text{-Co}_3\text{S}_4@\text{NF}$ electrode has a larger electric-field-induced locally adsorbed S^{2-} concentration (*Nature*, 2016, **537**, 382-386.). This absorbing test result is consistent with the simulation results.

The electrical properties of as-prepared catalysts were further evaluated by KPFM (*Nature*, 2016, **537**, 382-386.). The results also confirmed that the electric field is higher for the $n\text{-Co}_3\text{S}_4@\text{NF}$ with needle-like structure than the $r\text{-Co}_3\text{S}_4@\text{NF}$ with rod-like structure (**Fig. R4**). These results are consistent with the simulation results.

These experimental evidences indicate that the tip structure plays a crucial role in improving the attraction and mass transfer rate of ions in this region. It is predicted that the Co_3S_4 with a needle-like structure can exhibit excellent SOR activity.

The added figures and discussion have been provided in **Supplementary Figs. 12 and 14-16** in the revised SI and MS, respectively. The details for the finite element numerical calculation have been provided in the experimental section of the revised SI.

The added figures (see **Figs. R1-4**), discussion in the revised MS and details for the finite element numerical calculation in the revised SI are also provided as follows for you to review.

“Kelvin probe atomic force microscopy experimentally confirmed that the electric field is higher for the $n\text{-Co}_3\text{S}_4@\text{NF}$ with needle-like structure than the $r\text{-Co}_3\text{S}_4@\text{NF}$ with rod-like structure (Supplementary Fig. 12). We then experimentally evaluate the effect of tip structure in shaping the local environments. We performed the S^{2-} absorbing test by measuring the concentration of adsorbed S^{2-} on the electrode (Supplementary Fig. 14). The ultraviolet and visible (UV-vis) absorption peak of S^{2-} is

located at the wavelength of 230 nm. The S^{2-} absorption test shows that the number of S^{2-} adsorbed by the n-Co₃S₄@NF electrode is more than twice that of the r-Co₃S₄@NF electrode. The electrochemically active surface area (ECSA) was found to have a positive relationship with the double layer capacity (C_{dl}), but the C_{dl} value of the n-Co₃S₄@NF electrode is less than twice that of the r-Co₃S₄@NF electrode (Supplementary Figs. 15 and 16).^{26,29} This indicates that the n-Co₃S₄@NF electrode has a larger electric-field-induced locally absorbed S^{2-} concentration.¹⁷ This absorbing test result is consistent with the simulation results. It indicates that the tip structure plays a crucial role in improving the attraction and mass transfer rate of ions in this region. It is predicted that the Co₃S₄ with a needle-like structure can exhibit excellent SOR activity.” (Page 2 and 5 in revised MS)

“Kelvin probe force microscopy (KPFM) measurement

KPFM measurement is a powerful technique for simultaneous obtaining the topography and surface potential with nanometer scale spatial resolution and millivolt sensitivity in potential resolution via a dual pass process. The KPFM measurement was implemented based on an AFM (NT-MDT NTEGRA) system using a conductive NSG30/Pt tip. All the images were recorded in semi contact mode at room temperature with a relative humidity of 30%. Si cantilevers (NT-MDT with a typical curvature radius of the tip of 35 nm and a typical resonant frequency of 0.5 kHz) were used. In a standard KPFM procedure, a DC bias voltage is continually adjusted with a potential feedback loop to nullify the surface potential difference between the probe and sample. During the SemiContact 2-pass KPFM measurement, an AC bias voltage, superimposed on a DC bias voltage, is applied between the probe and sample. Thus, the potential difference can be written as

$$V = V_{dc} - \Delta V + V_{ac} \sin(\omega t)$$

where V_{dc} and V_{ac} are DC bias and AC bias, respectively. The $\Delta V = (\phi_{sample} - \phi_{tip})/e$ is the contact potential difference (CPD) between the probe and sample.

Finite element analysis (FEA) We used the COMSOL Multiphysics solver to simulate the current density distribution and the S^{2-} concentration distribution under the preset electric field in the electrolyte around the surface of different catalysts, which is

simplified in the conical shape to investigate the local situations.³ The dimensions of these systems are set at the scale of ten nanometers and one micrometer.

The current density distribution is simulated by the steady state research in the 'primary current distribution' module. The electrolyte conductivity was assumed to be 5 S m^{-1} . The potential losses due to electrode kinetics and mass transport are assumed to be negligible, and ohmic losses are governed by the current distribution in the cell. The current density was computed using the equation:

$$\nabla \cdot i_l = Q_l, \quad i_l = -\sigma_l \nabla \Phi_l,$$

where i_l represents the local current density in the electrolyte, σ_l represents the assumed electrolyte conductivity, and Φ_l represents the preset voltage of 50 mV.

The concentration distribution under the electric field migration around the catalyst surface is simulated by the transient state research in 'dilute material transfer' coupled with a 'primary current distribution' module, which was conducted to provide the preset electric field with the given voltage of 1.0 V. The S^{2-} concentration distribution was simulated by the equation:^{4,5}

$$\frac{\partial c_i}{\partial t} + \nabla \cdot J_i = R_i, \quad J_i = -D_i \nabla c_i - z_i u_{m,i} F c_i \nabla V,$$

where $\nabla \cdot J_i$ represents the contribution from the external factors, including the concentration gradient and ion migration in the electric field in this model. D_i represents the diffusion coefficient, which is set as $10^{-9} \text{ m}^2 \text{ s}^{-1}$ here. z_i is the ion charge of S^{2-} , and $u_{m,i}$ represents the ion mobility of $10^{-13} \text{ s mol kg}^{-1}$.” (Page 4-5 in revised SI)

Fig. R1 | S^{2-} absorbing test.

Fig. R2 | The CV curves of $n\text{-Co}_3\text{S}_4\text{@NF}$ and $r\text{-Co}_3\text{S}_4\text{@NF}$ under different scan rates in the non-Faraday region.

Fig. R3 | C_{dl} of catalysts derived from the current density versus the scan rate.

Fig. R4 | Topography images of (a) $n\text{-Co}_3\text{S}_4$ and (b) $r\text{-Co}_3\text{S}_4$; Surface potential images of (c) $n\text{-Co}_3\text{S}_4$ and (d) $r\text{-Co}_3\text{S}_4$. Contact potential difference of (e) $n\text{-Co}_3\text{S}_4$ and (f) $r\text{-Co}_3\text{S}_4$.

Original comment 1-2:

The citations need improvement. More relevant reference should be cited at proper place such as line 100, line 150, line 155, line 159, line 277... Also, are there any reference in which the local electric field and the ion concentration are analyzed by the finite element numerical calculation? If yes, relevant reference should be cited.

Response 1-2. Thank you very much for pointing out these issues. We have made appropriate corrections and highlighted the contents in yellow in the revised MS. Some references (including (*Energy Environ. Sci.* **16**, 285-294 (2023), *Nature* **537**, 382-386 (2016), *Adv. Mater.* **33**, e2007377 (2021)) which have analyzed the local electric field and the ion concentration by the finite element numerical calculation have been cited in the revised MS.

Line 100: “Given the local field enhancement effect of needle-like structures in improving the performance of electrocatalytic reactions, we attempt to apply this structure to the electrocatalytic SOR process (*Energy Environ. Sci.* **16**, 285-294 (2023), *Nature* **537**, 382-386 (2016), *Adv. Mater.* **33**, e2007377 (2021))”

Line 150: “It can enhance the adsorption of surrounding reactive ions (e.g., S²⁻ and OH⁻), further enhancing the SOR and HER catalytic activities. (*Energy Environ. Sci.* **16**, 285-294 (2023))”

Line 159: “The essence of this phenomenon is electrostatic repulsion, where free electrons migrate to the sharpest regions on the charged metal electrodes, which increases the free electron density and further leads to a climb in the local electric field strength. (*Nature* **537**, 382-386 (2016))”

Line 277: “As shown in Fig. 3f, the zeta potential of *n*-Co₃S₄@NF (-24.5 mV) compared to *n*-Co(OH)₂@NF (30.4 mV) is shifted negatively. This suggests that the *n*-Co₃S₄@NF catalyst provides a weaker binding energy to Cl⁻ and repels it. (*Adv. Funct. Mater.* **33**, 2212183 (2022))”

The relevant content in the revised MS is also provided as follows for you to review. “Given the local field enhancement effect of needle-like structures in improving the performance of electrocatalytic reactions, we attempt to apply this structure to the electrocatalytic SOR process.^{16,17,18} It can enhance the adsorption of surrounding

reactive ions (e.g., S^{2-} and OH^-), further enhancing the SOR and HER catalytic activities.¹⁶ It can be noticed that the current density in the sharpest region of the tip of the needle-like structure increases significantly, which is accompanied by enhanced electric field line densities in the localized region (Fig. 1g, h). The essence of this phenomenon is electrostatic repulsion, where free electrons migrate to the sharpest regions on the charged metal electrodes, which increases the free electron density and further leads to a climb in the local electric field strength.¹⁷ As shown in Fig. 3f, the zeta potential of $n-Co_3S_4@NF$ (-24.5 mV) compared to $n-Co(OH)_2@NF$ (30.4 mV) is shifted negatively. This suggests that the $n-Co_3S_4@NF$ catalyst provides a weaker binding energy to Cl^- and repels it.¹⁵” (Page 3, 4, 5, 8 and 9 in revised MS)

Original comment 1-3:

The analysis on the S 2p XPS spectra is not reasonable (line 138-141). In figure S9b and S10b, the peak at around 169 eV indicates the presence of SO_4^{2-} species. Also, it is not necessary to include Co-S peaks in the S 2p spectra. They are generally included in the S 2p_{1/2} and S 2p_{3/2} peaks.

Response 1-3. Thank you very much for pointing out these issues. The related XPS spectra have been re-deconvoluted (see **Figs. R5-7**).

The corresponding S 2p XPS spectra have been provided in **Supplementary Figs. 9b and 10b** in the revised SI.

The revised figures (see **Figs. R5 and R6**) and discussion in the revised MS are also provided as follows for you to review.

“The peaks at binding energies of 168.89, 164.00, and 161.83 eV are attributed to SO_4^{2-} , S 2p_{1/2}, and S 2p_{3/2}, respectively (Supplementary Figs. 9b and 10b)^{24,25}” (Page 4 in revised MS)

Fig. R5 | High-resolution S 2p XPS spectrum of *n*-Co₃S₄@NF.

Fig. R6 | High-resolution S 2p XPS spectrum of *r*-Co₃S₄@NF.

Original comment 1-4:

In the Co 2p XPS spectra, why the peak shift of *n*-Co₃S₄@NF is more obvious than *r*-Co₃S₄@NF?

Response 1-4. Thank you for your question. The related XPS spectra have been re-deconvoluted. The peak shift of *n*-Co₃S₄@NF is 0.29 eV. The peak shift of *r*-Co₃S₄@NF is 0.02 eV. The difference in binding energy shift may be due to significant differences in sample morphology. The similar phenomenon has also been reported in previous work (*Angew. Chem. Int. Ed.*, 2016, **55**, 9548-9552).

The Co 2p XPS spectra have been provided in **Supplementary Figs. 9a and 10a** in the revised SI, respectively.

The added figures (see **Figs. R7 and R8**) and discussion in the revised MS are also provided as follows for you to review.

“As shown in Supplementary Fig. 9a, the high-resolution Co 2p of the n-Co₃S₄@NF catalyst can be deconvoluted into two pairs of spin-orbit double peaks (Co 2p_{3/2} and Co 2p_{1/2}) with satellite peaks (Sat.). The peaks at 798.30 and 782.20 eV correspond to Co²⁺ and the additional two characteristic peaks at 796.70 and 781.46 eV correspond to Co³⁺.^{21,22} The characteristic peak of Co²⁺ exhibits a leftward shift of 0.29 eV relative to the precursor n-Co(OH)₂@NF. There is a shift caused by the dual contribution of the electronegativity and easy polarizability of the sulfion in Co₃S₄, which contributes to the adjustment of the charge distribution of the adjacent metal ions by reinforcing the electron transfer capacity.²³ Noteworthy, the Co²⁺ characteristic peak shift attributed to n-Co₃S₄@NF is more obvious in comparison with r-Co₃S₄@NF (Supplementary Figs. 9a and 10a), whereby we speculate that the special needle-like structure can facilitate the improvement of the catalytic ability of the Co₃S₄ catalyst.”

(Page 4 in revised MS)

Fig. R7 | High-resolution Co 2p XPS spectrum of n-Co(OH)₂@NF and n-Co₃S₄@NF.

Fig. R8 | High-resolution Co 2p XPS spectrum of $r\text{-Co(OH)}_2\text{@NF}$ and $r\text{-Co}_3\text{S}_4\text{@NF}$.

Original comment 1-5:

The adsorption of reactants such as OH^- , S^{2-} , Cl^- on catalyst surface is not solely determined by the electrostatic interaction. The zeta potential values cannot be used as the indicator for the interaction between Cl^- and catalyst (line 276).

Response 1-5. Thank you very much for pointing out this issue. The corrosive nature of seawater owing to the presence of Cl^- poses a greater challenge to the stability of electrocatalysts for hydrogen production by seawater electrolysis. We have provided more experimental evidences for the $n\text{-Co}_3\text{S}_4\text{@NF}$'s resistance to the Cl^- corrosion, including open circuit potential (OCP) test (**Fig. R9**), Tafel plot test (**Fig. R10**), LSV curves before and after the CP test (**Fig. R11**), and ICP-MS test for the dissolution rate of Co element in the electrolyte of 1 M NaOH seawater (**Fig. R12**). These results, along with zeta potential (**Fig. R13**), SEM (**Fig. R14**), and three-dimensional X-ray tomography (**Fig. R15**) results prove that the $n\text{-Co}_3\text{S}_4\text{@NF}$ is suitable for long-term electrocatalytic HER due to its resistance to the Cl^- corrosion.

The details for the added discussion and related figures have been provided in the revised MS and SI (**Fig. 3f, Supplementary Figs. 20 and 28-32**).

The related figures (see **Figs. R9-15**) and discussion in the revised MS are also provided as follows for you to review.

“As shown in Fig. 3f, the zeta potential of $n\text{-Co}_3\text{S}_4\text{@NF}$ (-24.5 mV) compared to $n\text{-Co(OH)}_2\text{@NF}$ (30.4 mV) is shifted negatively. This suggests that the $n\text{-Co}_3\text{S}_4\text{@NF}$ catalyst provides a weaker binding energy to Cl^- and repels it.¹⁵ Tafel plots show the corrosion potential of $n\text{-Co}_3\text{S}_4\text{@NF}$ is more positive than that of Ni foam in a 1 M NaOH seawater electrolyte (Supplementary Fig. 28), indicating a lower corrosion tendency and the improved corrosion resistance of the $n\text{-Co}_3\text{S}_4$ electrocatalyst compared to a blank Ni foam substrate.³³ By testing the change of open circuit potential (OCP) along the CP test, the $n\text{-Co}_3\text{S}_4$ with high corrosion resistance shows negligible degradation over time (Supplementary Fig. 29). LSV curves before and after the CP test show a small degree of degradation and confirm the durability of the $n\text{-Co}_3\text{S}_4\text{@NF}$ (Supplementary Fig. 30). As shown in Supplementary Fig. 31, the dissolution rate of Co element in the electrolyte of 1 M NaOH seawater from the $n\text{-Co}_3\text{S}_4\text{@NF}$ is only about 10-20 μM , which is quite negligible. Obviously, the $n\text{-Co}_3\text{S}_4\text{@NF}$ is suitable for long-term electrocatalytic HER due to its resistance to Cl^- corrosion. Moreover, the Faraday efficiency of $n\text{-Co}_3\text{S}_4\text{@NF}$ up to 99.1% reveals that no additional side reactions occur during hydrogen production in alkaline seawater, highlighting the high efficiency of the catalyst in HER. (Fig. 3h). Its backbone structure also retained substantial integrity upon the durability test, also indicating that the $n\text{-Co}_3\text{S}_4\text{@NF}$ catalyst has excellent resistance to seawater corrosion (Supplementary Fig. 32).”

“Simultaneously, we examined the crystal structures, morphology, and surface chemical states of the $n\text{-Co}_3\text{S}_4\text{@NF}$ catalyst after the HER stability test in the alkaline seawater electrolyte (Supplementary Figs. 19-22). The $n\text{-Co}_3\text{S}_4\text{@NF}$ catalyst showed negligible changes, presenting the vast potential of the $n\text{-Co}_3\text{S}_4\text{@NF}$ catalyst for hydrogen production by seawater electrolysis.” (Pages 9 and 10 in revised MS)

Fig. R9 | Open circuit potential (OCP) of $n\text{-Co}_3\text{S}_4\text{@NF}$ before and after 10 h stability test in 1 M NaOH seawater at a current density of 100 mA cm^{-2} .

Fig. R10 | Tafel plots of $n\text{-Co}_3\text{S}_4\text{@NF}$ and Ni foam in a 1 M NaOH seawater electrolyte.

Fig.R11 | LSV curves of the $n\text{-Co}_3\text{S}_4\text{@NF}$ catalyst before and after the 210 h CP test in 1 M NaOH seawater.

Fig. R12 | Atomic concentrations of Co in the electrolyte during the chronopotentiometry test at the current density of 100 mA cm^{-2} for 150 h.

Fig. R13 | Zeta potentials of $n\text{-Co(OH)}_2\text{@NF}$ and $n\text{-Co}_3\text{S}_4\text{@NF}$.

Fig. R14 | SEM images of $n\text{-Co}_3\text{S}_4\text{@NF}$ (a) before (b) after HER stability test in 1 M NaOH seawater.

Fig. R15 | Three-dimensional X-ray tomography images of $n\text{-Co}_3\text{S}_4\text{@NF}$ electrodes (a) before and (b) after 210 h HER stability test at the current density of 100 mA cm⁻² in 1 M NaOH seawater.

Original comment 1-6:

The sulfion concentration (1M) used in the electrolyzer is too high. What is the sulfion concentration of typical sulfion sewage? The authors are encouraged to collect industrial sulfion sewage and used in the electrolyzer so as to provide a more practical demonstration.

Response 1-6. We thank for the reviewer's valuable questions. We have collected the realistic sulfion-containing sewage from a natural gas field located in Dazhou, Sichuan Province, PRC (**Fig. R16a**). The concentration of sulfion is lower than 1 M (**Fig. R16b**). Due to strict environmental regulations and safety considerations, we can only collect diluted sulfur-containing wastewater from natural gas fields, resulting in a low concentration of sulfion. Therefore, when the obtained realistic sulfion-containing sewage is directly used as the SOR electrolyte, the electrocatalyst exhibits weak SOR catalytic performance (**Fig. R16c**). Furthermore, we also attempted to construct a novel HSE demonstration with $n\text{-Co}_3\text{S}_4\text{@NF}$ acting as both anode and cathode for electrocatalytically selective removal of H_2S from simulated industrial syngas (49% CO , 49% H_2 , and 2% H_2S). It is worth noting that the concentration of H_2S in the simulated industrial syngas is comparable to that of the simulated sulfion wastewater (1 M Na_2S + 1 M NaOH) used in our work (**Fig. R17**). This device exhibits a high response current and a long durability of more than 120 h (**Figs. R18 and R19**), which is important in terms of selective removal of H_2S from industrial syngas and energy-efficient hydrogen production. On this basis, by using seawater and industrial sulfion-containing wastewater into renewable energy power HSE, the burden of hydrogen production on electrical grids can be removed and sustainable hydrogen production scale can be expanded.

The details for the related content have been provided in **Supplementary Figs. 24 and 36-38** in the revised MS and SI.

The added figures (see **Figs. R16-19**) and discussion in the revised MS are also provided as follows for you to review.

“To determine the effectiveness of $n\text{-Co}_3\text{S}_4@\text{NF}$ in treating realistic sulfion-containing sewage by electrocatalytic SOR method. The realistic sulfion-containing sewage collected from a natural gas field located in Dazhou, Sichuan Province, PRC (Supplementary Fig. 24a). The realistic sulfion-containing sewage is directly used as the SOR electrolyte, in which the $n\text{-Co}_3\text{S}_4@\text{NF}$ electrocatalyst requires a potential of 486 mV vs. RHE to achieve a current density of 100 mA cm⁻². And the $n\text{-Co}_3\text{S}_4@\text{NF}$ electrocatalyst can only achieve a current density of 150 mA cm⁻² at the potential of 500 mV vs. RHE. Due to strict environmental regulations and safety considerations, we can only collect diluted sulfur-containing wastewater from natural gas fields, resulting in a low concentration of sulfion. The pH value of the solution and the concentration of sulfion are much lower than the simulated sulfion wastewater configured in the laboratory, therefore $n\text{-Co}_3\text{S}_4$ exhibits weak catalytic performance (Supplementary Fig. 24b, c). Due to the realistic sulfion-containing wastewater we collected from the natural gas field has undergone dilution and is not suitable for direct use as an electrolyte for SOR (Supplementary Fig. 24). To determine the effectiveness of $n\text{-Co}_3\text{S}_4@\text{NF}$ in SOR-assisted seawater electrolysis hydrogen production and sulfion-containing sewage purification in a realistic system, we constructed a demo HSE, in which $n\text{-Co}_3\text{S}_4@\text{NF}$ is used as both anode and cathode, for the electrocatalytic selective removal of H₂S from simulated industrial syngas (49% CO, 49% H₂, and 2% H₂S).⁵⁰ As shown in Supplementary Fig. 36, UV-vis spectra show that the sulfion’s concentration of 2% H₂S/syngas + 1 M NaOH is comparable to that of the simulated sulfion-containing wastewater (1 M Na₂S + 1 M NaOH). The LSV test showed that the HSE using 2% H₂S/syngas + 1 M NaOH as anolyte had a high response current density, which is comparable to that of the HSE using 1 M Na₂S + 1 M NaOH as anolyte (Supplementary Fig. 37). Furthermore, the long-term galvanostatic test at 100 mA cm⁻² current density showed that the HSE using 2% H₂S/syngas + 1 M NaOH as anolyte could be well maintained for over 120 h (Supplementary Fig. 38). This suggests its excellent stability in the selective removal of H₂S from industrial syngas and energy-saving hydrogen production.” (Page 7, 8, 13 and 14 in revised MS)

Fig. R16 | A demo for the $n\text{-Co}_3\text{S}_4@\text{NF}$ used as SOR electrocatalyst for the purification of realistic sulfion-containing sewage from a natural gas field. (a) The optical photograph of the sulfion-containing sewage collected from Dazhou, Sichuan Province, PRC. (b) UV-vis spectra of different solutions. (c) The LSV curves for the SOR over $n\text{-Co}_3\text{S}_4@\text{NF}$ in different reaction solutions, including the collected realistic sulfion-containing sewage and 1 M Na_2S + 1 M NaOH as a blank control.

Fig. R17 | UV-vis spectra for 2% H_2S /syngas and 1 M Na_2S + 1 M NaOH .

Fig. R18 | SOR polarization curves of $n\text{-Co}_3\text{S}_4@\text{NF}$ in different electrolytes.

Fig. R19 | Durability measurement of $n\text{-Co}_3\text{S}_4@\text{NF}$ for removing H_2S in industrial syngas via 1 M NaOH solution with 2% H_2S /syngas. The measurement was carried out at a galvanostatic current of 100 mA cm^{-2} , and the corresponding potential of $n\text{-Co}_3\text{S}_4@\text{NF}$ was maintained at around 0.8 V for 120 h. The fresh electrolytes were changed every 24 h.

Response to the Reviewer #2

General Comments:

This manuscript fabricated a bifunctional $n\text{-Co}_3\text{S}_4@\text{NF}$ catalyst with a needle-like structure for SOR-assisted energy-saving hydrogen production by seawater electrolysis. The HSE system assembled by the bifunctional $n\text{-Co}_3\text{S}_4@\text{NF}$ is not only able to reduce power consumption up to 67.9% compared to the conventional ASE system but can

also be grid-connected to a thermoelectric/photovoltaic power generation system for efficient and long-lasting hydrogen production. The results from multiple characterizations and calculations are reliable. Besides, the plots and figures are well-designed and shown in a logical manner. Overall, this work is interesting and this manuscript is well organized, I would be pleased to recommend that it be published after minor revision. Detailed comments are listed as follows:

Response. Thank you for your very positive comment on our work and your recommendation for publication after revision.

Original comment 2-1:

Regarding the DFT calculation, structural models of reaction intermediates adsorbed on the catalyst should be provided.

Response 2-1. Thank you for your suggestion. Structural models of reaction intermediates adsorbed on the catalyst have been provided in **Supplementary Fig. S34** in the revised SI. The figures are also provided as follows for you to review (see **Fig. R20**).

Fig. R20. | Structural models of reaction intermediates adsorbed on the (a) (311) facet, (b) (100) facet, (c) (111) facet, and (d) (110) facet of Co₃S₄.

Original comment 2-2:

The details for Comsol simulations should be provided.

Response 2-2. Thank you for your suggestion. The details for Comsol simulations have been provided in the revised SI. The corresponding details are also provided as follows for you to review.

“Finite element analysis (FEA) We used the COMSOL Multiphysics solver to simulate the current density distribution and the S²⁻ concentration distribution under the preset electric field in the electrolyte around the surface of different catalysts, which is simplified in the conical shape to investigate the local situations.³ The dimensions of these systems are set at the scale of ten nanometers and one micrometer.

The current density distribution is simulated by the steady state research in the ‘primary current distribution’ module. The electrolyte conductivity was assumed to be 5 S m⁻¹. The potential losses due to electrode kinetics and mass transport are assumed to be negligible, and ohmic losses are governed by the current distribution in the cell. The current density was computed using the equation:

$$\nabla \cdot i_l = Q_l, \quad i_l = -\sigma_l \nabla \Phi_l,$$

where i_l represents the local current density in the electrolyte, σ_l represents the assumed electrolyte conductivity, and Φ_l represents the preset voltage of 50 mV.

The concentration distribution under the electric field migration around the catalyst surface is simulated by the transient state research in ‘dilute material transfer’ coupled with a ‘primary current distribution’ module, which was conducted to provide the preset electric field with the given voltage of 1.0 V. The S²⁻ concentration distribution was simulated by the equation:^{4,5}

$$\frac{\partial c_i}{\partial t} + \nabla \cdot J_i = R_i, \quad J_i = -D_i \nabla c_i - z_i u_{m,i} F c_i \nabla V,$$

where $\nabla \cdot J_i$ represents the contribution from the external factors, including the concentration gradient and ion migration in the electric field in this model. D_i represents the diffusion coefficient, which is set as 10⁻⁹ m² s⁻¹ here. z_i is the ion charge of S²⁻, and $u_{m,i}$ represents the ion mobility of 10⁻¹³ s mol kg⁻¹.” (Page 4 in revised SI)

Original comment 2-3:

The authors have conducted sufficient theoretical and experimental research on the promotion of SOR performance by needle-like structure design of catalyst. Is the needle-like structure design of the catalyst beneficial for HER performance? Please add more discussions about it.

Response 2-3. Thank you for your question. The needle-like structure design of the catalyst beneficial for its HER performance. This architecture with a high curvature leads to the accumulation of electrons around the needle regions, which induces a local electric field and accelerates the HER catalytic kinetics. This phenomenon has also been reported in other literature (*Nat. Energy*, 2019, **4**, 512-518).

Original comment 2-4:

The annotation text and scale in the figure should be clear. For example, Fig. 3d, Fig S19b, and Fig. 23.

Response 2-4. Thank you very much for pointing out these issues. We have made appropriate corrections in the revised manuscript.

Original comment 2-5:

The authors are suggested to explain why there appears fluctuations in durability test of the HSE and ASE systems in Figure 4d..

Response 2-5. Thank you very much for pointing out these issues. The long-term durability of the HSE hydrogen production system is showed in Fig. 4d and the electrolyte was changed every 24 h. It was observed that the potential gradually increased as the S^{2-} concentration in the solution decreased. However, the potential was maintained at about 0.7 V (vs. RHE) even after 21 times electrolyte changes, indicating its excellent stability for SOR.

Original comment 2-6:

As SOR can proceed in different ways, one may be curious why electrochemical reaction ($S^{2-} - 2e^- \rightarrow S$) or products (S) are dominated in the present work, and how the authors regulated no further oxidation happens on S.

Response 2-6. Thank you for your question. The prevalent oxidation of S^{2-} can occur in three different stages, i.e., an initial S^{2-} state, an intermediate S_x^{2-} state, and S as the final product. Due to the low applied potential of our SOR electrolysis system, S will not be further oxidized to higher valence (*Water Res.*, 2008, **42**, 4965–4975.). At much higher potentials, S can be oxidized to sulfite or sulphate. For example, I. C. HAMILTON reported initial S^{2-} can be oxidized to S at 0.1 V vs. SHE, and sulphur oxidation occurs at 1.25 V vs. SHE (*J. Appl. Electrochem.*, 1983, **13**, 783–794). In our work, the *n*-Co₃S₄@NF electrode requires extremely low potentials of 233 and 279 mV (vs. RHE) to afford SOR current densities of 100 and 300 mA cm⁻², respectively. The applied potential is much lower than the potential needed for sulfur's oxidation.

Response to the Reviewer #3**General Comments:**

In this manuscript, the authors design a needle-like Co₃S₄ bifunctional catalyst grown on nickel foam (NF) with a unique tip structure (*n*-Co₃S₄@NF), which can achieve energy-saving hydrogen production, sulfion-rich wastewater purification, and sulfur recovery. Both experiments and calculations have been applied to explain the performance. This study is novel and very interesting. The data and methodology are reliable and the manuscript is also well organized and presented clearly. The conclusion is also clear and reliable. This manuscript can be considered for publication after addressing the following minor issues.

Response. Thank you for your very positive comment on our work and your recommendation for publication after revision.

Original comment 3-1:

The HSE system rate by the 1.0 V commercial solar cell should provide a video.

Response 3-1. Thank you for your suggestion. The HSE system rate by the 1.0 V commercial solar cell have been provided in the revised Supplementary Movie.

Original comment 3-2:

There are some typos in the manuscript, which should be corrected. For example, "but is far higher than that of n-Co(OH)₂@NF and r-Co(OH)₂@NF"; in Figure 3e and Figure 3i, "vs. RHE".

Response 3-2. Thank you very much for pointing out these issues. We have made appropriate corrections and highlighted the contents in yellow in the revised manuscript in the revised manuscript.

Original comment 3-3:

Figure 4g is slightly blurry and should be improved appropriately.

Response 3-3. Thank you very much for pointing out this issue. We have made appropriate corrections. The updated figures are also provided as following for you to review (see in **Fig. R21**).

Fig. R21 | A rough techno-economic analysis of the HSE and ASE systems for hydrogen production.

Original comment 3-4:

The electro-oxidation products (elemental sulfur) of the sulfur oxidation reaction are prone to passivating the electrode surface and increasing the reaction overpotential, even making continuous operation infeasible. Can the design of the needle structure weaken this passivation effect?

Response 3-4. Thank you for your question. We believe that the design of the needle structure plays a role in weakening this passivation effect. First, from a geometric perspective, the needle structure has a smaller scale, making it difficult for sulfur to adsorb on the catalyst surface or more easily desorb, thereby avoiding the formation of a thick sulfur adsorption layer. Second, materials with needle structures have a larger electrochemical active area, and the generated sulfur does not easily cover all surfaces.

REVIEWERS' COMMENTS

Reviewer #2 (Remarks to the Author):

The authors did a good job to improve the manuscript, and all the concerns raised were well addressed, the revision is thus recommended to be accepted for Nat Commun Publication as it was.

Reviewer #3 (Remarks to the Author):

The revisions on the manuscript is sound and I am happy to suggest its acceptance in Nat Commun. During the proofing process, it would be better if more classical or timing reports on related topic could be cited, and the English writing could be further improved.

Point-by-point response to the reviewers' comments

General response:

We appreciate all the reviewers' professional comments and suggestions, with which we have improved the quality of our manuscript.

Reviewer #2 (Remarks to the Author):

The authors did a good job to improve the manuscript, and all the concerns raised were well addressed, the revision is thus recommended to be accepted for Nat Commun Publication as it was.

Response:

We thank this reviewer for his/her comments here! We particularly appreciate his/her positive comment.

Reviewer #3 (Remarks to the Author):

The revisions on the manuscript is sound and I am happy to suggest its acceptance in Nat Commun. During the proofing process, it would be better if more classical or timing reports on related topic could be cited, and the English writing could be further improved.

Response. Thank you for your very positive comment on our work and your recommendation for publication after revision. The corresponding details are also provided as follows for you to review.

“H₂ production methods suffer from severe environmental pollution, complex equipment processes, high investment operational costs, and excessive energy consumption.³ In contrast, electrocatalytic water splitting has received widespread attention as a zero-pollution emission and efficient method to produce H₂.^{4,5} As the half-reaction of water electrolysis, the anodic OER is a four-electron transfer process with

*sluggish kinetics, leading to increased overall energy consumption.*¹⁰

3. Wang T, et al. Combined anodic and cathodic hydrogen production from aldehyde oxidation and hydrogen evolution reaction. *Nat. Catal.* 5, 66-73 (2021).

5. Xie H, et al. A membrane-based seawater electrolyser for hydrogen generation. *Nature* 612, 673–678 (2022).

10. Liu Y, et al. Manipulating dehydrogenation kinetics through dual-doping Co_3N electrode enables highly efficient hydrazine oxidation assisting self-powered H_2 production. *Nat. Commun.* 11, 1853 (2020).

” (Page 1, 2 and 19 in revised Manuscript)